# Simultaneously tuning interlayer spacing and termination of MXenes by Lewis-basic halides

Tianze Zhang [1,2], Libo Chang [1], Xiaofeng Zhang[3], Hujie Wan [1], Na Liu [1], Liujiang Zhou [2,3] & Xu Xiao [1,2] ✉

The surface and interface chemistry are of significance on controlling the properties of two-dimensional transition metal carbides and nitrides (MXenes). Numerous efforts have been devoted to the regulation of $Ti_3C_2T_x$ MXene, however, tuning interlayer spacing and surface halogen termination of other MXenes (besides $Ti_3C_2T_x$) is rarely reported while demanded. Here we propose a Lewis-basic halides treatment, which is capable of simultaneously engineering the interlayer spacing and surface termination of various MXenes. Benefited from the abundant desolvated halogen anions and cations in molten state Lewis-basic halides, the -F termination was substituted by nucleophilic reaction and the interlayer spacing was enlarged. $Ti_3C_2T_x$ MXene treated by this method showed a high specific capacity of 229 mAh g$^{-1}$ for Li$^+$ storage, which is almost 2 times higher than pristine one. Considering the universality, our method provides an approach to regulating the properties of MXenes, which may expand their potential applications in energy storage, optoelectronics and beyond.

As the emerging member of two-dimensional (2D) materials, 2D transition metal carbides and nitrides (MXenes) have been attracting great attention in energy conversion/storage[1–3], optoelectronics[4], electromagnetic interference shielding[5], etc., due to their metallic conductivity and processability[6]. MXenes have a general formula written as $M_{n+1}X_nT_x$ ($n = 1–4$), where M represents early transition metal elements (Ti, Nb, Mo, V, W, etc.), X is carbon and/or nitrogen, and $T_x$ stands for the surface terminations[7–9]. Notably, terminations and interlayer spacing are of importance on tuning the physical and chemical properties of MXenes[10]. For instance, larger interlayer spacing is beneficial to more and fast ion transport, resulting in high rate capability for energy storage[11,12]. Chlorine terminated MXene showed much higher capacity as the electrode material for Li$^+$ storage[13], while fluorine terminated MXene hinders the Li$^+$ transport[14,15]. The controllable superconductivity in $Nb_2CT_x$ by tuning terminations was also reported[16]. The interlayer spacing and termination species of MXenes are mainly determined by the synthesis method.

In general, MXenes are obtained by selectively etching A element from precursor MAX (A is a main group element mainly including Al, Ga, Si, etc.) in hydrofluoric acid (HF) containing solution or Lewis acid molten salt (the product is denoted as multilayer MXene)[17,18]. For HF solution method, various and content-uncontrollable terminations, such as -O, -OH and -F are inevitable and predominant on the surface of MXenes[19–21]. In addition, although the interlayer spacing (d-spacing of (002) facet) of MXene is larger than MAX after etching, it is still pretty small in the range of about 9.5–11.2 Å[22,23]. To further obtain monolayer MXene or use multilayer MXene as electrode material, enlarging interlayer spacing is necessary which is always conducted with the assistance of intercalants. Unfortunately, except for $Ti_3C_2T_x$ whose interlayer spacing could be enlarged by neutral inorganic lithium salt (such as LiCl), other MXenes always need to be intercalated by inorganic base[24] (LiOH, NaOH) or organic base[25] (such as tetramethylammonium hydroxide, TMAOH), which may destroy the

[1]School of Electronic Science and Engineering, University of Electronic Science and Technology of China, Chengdu, Sichuan 611731, China. [2]Yangtze Delta Region Institute (Huzhou), University of Electronic Science and Technology of China, Huzhou, Zhejiang 313001, China. [3]School of Physics, University of Electronic Science and Technology of China, Chengdu 611731, China. ✉e-mail: xuxiao@uestc.edu.cn

structure of MXene and induce defects[26]. This always leads to low conductivities (for instance, $Ti_3C_2T_x$ freestanding film shows a high conductivity up to 24,000 S cm$^{-1}$ if delaminated by LiCl[27], while less than 10 S cm$^{-1}$ is achieved for the ones delaminated by LiOH and NaOH[22]). For Lewis acid molten salt method (above 450 °C), termination is more controllable and tunable[28]. For instance, Cl-terminated MXenes followed by S-, Te-, O- (etc.) termination substitution are achieved[16]. However, in molten salt system, the resulted MXenes showed a relatively small interlayer spacing (about 10.9 Å), in which the interlayer spacing could only be enlarged by organic reagents (difficult to be removed) intercalation in previous researches[16,29]. Accordingly, pursuing general route to tuning the interlayer spacing and termination of various MXenes is of great significance while still keep challenging, not to mention developing a method that is capable of simultaneously realizing the regulation.

Ions are involved in intercalation process and termination substitution, through ion exchange[30] (with cations) and nucleophilic reaction[16] (with anions), respectively. If the above termination substitution process occurs in aqueous solution, the overall activation energy increases due to the energy consumption for the necessary step of ion de-solvation, which limits the reaction rate according to the Arrhenius equation[31]. At the same time, the desolvated cations intercalate between MXene layers to balance charge after surface termination substitution. In this context, the molten salt system is of particular interest as "naked" anions are beneficial for termination substitution and "naked" cations could simultaneously intercalate MXenes. However, the melting point of most individual ionic crystals is above 400 °C, which may be not suitable for MXene as self-oxidation of MXene occurs at high temperature even in argon[32]. Notably, eutectic molten salts, which consists of several different salts, exhibit a much lower melting point than each component[33]. Interestingly, previous studies have shown that adjusting the proportion of each component in eutectic molten salts can realize the transition between Lewis acidity and Lewis basicity[33,34]. For instance, in $AlCl_3$/NaCl/KCl molten salt system (melting point is about 90 °C), when the molar ratio of $AlCl_3$ is higher than 50%, the molten system shows Lewis acidity, mainly containing $[AlCl_4]^-$, $[Al_2Cl_7]^-$, $Al_2Cl_6$ and $AlCl_3$, with less Na$^+$ and K$^+$. In contrast, when the molar ratio of $AlCl_3$ is less than 50%, the molten system shows Lewis basicity with the presence of abundant Na$^+$, K$^+$, Cl$^-$, along with $[AlCl_4]^-$ and $[Al_2Cl_7]^-$. Considering the relatively low melting point and abundant desolvated cations and halogen anions, we propose that utilizing the low-temperature Lewis-basic eutectic molten salt system may be an effective route to simultaneously tuning interlayer spacing and surface termination of MXene.

Herein, we report a general method to simultaneously engineering the surface termination and enlarging the interlayer spacing of MXenes by Lewis-basic halides. Lewis-basic $AlBr_3$/NaBr/KBr and $AlI_3$/NaI/KI eutectic molten salts with melting points of about 90 and 155 °C, respectively, are selected for MXene treatment (the treated samples are denoted as LB-MXene). As shown in Fig. 1a, the interlayer spacing of MXene could be enlarged due to the synergism of termination substitution and desolvated cations intercalation (Na$^+$ and K$^+$) in Lewis-basic halides, and the -F termination is substituted through reacting with desolvated Br$^-$ and I$^-$ (Fig. 1b). Interestingly, this method is generally applicable to various MXenes, including mono/double metal MXenes, 32 and 43 phase MXenes, and MXenes synthesized from different methods. The potential application of using LB-$Ti_3C_2T_x$ as electrode material for Li$^+$ storage is demonstrated, which shows capacity almost two times higher than pristine $Ti_3C_2T_x$.

## Results and discussion

### Preparation and characterization of LB-$Ti_3C_2T_x$

As shown in Fig. 1c, after HF etching $Ti_3AlC_2$ MAX phase, as-obtained multilayer $Ti_3C_2T_x$ consists of stacked $Ti_3C_2T_x$ flakes with interlayer spacing of 11.2 Å according to the X-ray diffraction (XRD) pattern

(Fig. 1e). $AlBr_3$/NaBr/KBr eutectic molten salt was chose to treat $Ti_3C_2T_x$, which has a molar ratio of 48.7:15.2:36.1 to form a Lewis-basic bromide. After treatment, the LB-$Ti_3C_2T_x$ shows an obvious expanded layer structure (Fig. 1d), in which the interlayer spacing increases to 14.7 Å (6.01° in XRD, Fig. 1e). Such a large interlayer spacing is superior to that of many reported value of MXenes intercalated by various cation species[35,36]. The interlayer spacing data of MXenes are summarized in Supplementary Table 1. Utilizing the large interlayer spacing, the monolayer LB-$Ti_3C_2T_x$ nanosheets (Supplementary Fig. 1a) can be obtained by further delamination of multilayer LB-$Ti_3C_2T_x$, forming a stable colloidal solution (Supplementary Fig. 2a). The in-plane crystal structure of monolayer LB-$Ti_3C_2T_x$ was checked by selected area electron diffraction pattern (SAED) in transmission electron microscopy (TEM). Typical hexagonal structure with similar interplanar crystal spacings compared to reported literatures is shown in Supplementary Fig. 1b[37], indicating that the Lewis-basic bromide treatment has no effect on changing the in-plane crystal structure of $Ti_3C_2T_x$.

The specific atomic arrangement of LB-$Ti_3C_2T_x$ along c-axis is performed by scanning transmission electron microscopy (STEM). As shown in Fig. 1f, five layers of atoms are clearly observed for each LB-$Ti_3C_2T_x$ monolayer. The middle bright three layers are Ti atoms, and the atoms on both sides are surface termination. In order to verify the species of surface termination, energy dispersive spectrum (EDS) line scanning was performed. The difference of line scans of Br and Ti elements validate the presence of -Br termination on the surface of LB-$Ti_3C_2T_x$ (Fig. 1g), with further validation by the even distribution of Br element in LB-$Ti_3C_2T_x$ from the element mapping in SEM as shown in Supplementary Fig. 3c. X-ray photoelectron spectroscopy (XPS) was used to further reveal the species of surface terminations. As shown in Supplementary Figs. 4 and 5 (Supplementary Table 2 for fitting details), the peaks of Br appear in the XPS survey spectrum of LB-$Ti_3C_2T_x$, while absence in pristine $Ti_3C_2T_x$. The high-resolution Br 3d region could be split into two Ti-Br bonds (Fig. 1h), indicating the covalent bonding between Br and Ti instead of simple adsorption occurs on the surface of LB-$Ti_3C_2T_x$[37]. In addition, the intensity of F decreases significantly (Supplementary Fig. 6).

### Mechanism of Lewis-basic halides treatment

In order to unveil the mechanism of simultaneously tuning interlayer spacing and surface terminations of $Ti_3C_2T_x$ by $AlBr_3$/NaBr/KBr eutectic molten salt, we treated $Ti_3C_2T_x$ in molten state $AlBr_3$/NaBr/KBr with different molar ratio (with molar percentage of $AlBr_3$ being 100, 79.9, 60.1 and 48.7 mol%, denoted as 100, 79.9, 60.1 and 48.7 mol% $AlBr_3$, the photographs of $AlBr_3$/NaBr/KBr molten salts are shown in Supplementary Fig. 7). As we mentioned previously, the molar ratio of each component is of great importance on controlling the chemical property of their eutectic molten salt[34]. Since $AlBr_3$ is molecular crystal, the molten state of pure $AlBr_3$ (100 mol%) does not contain any ions (as shown in Fig. 2a). Adding NaBr/KBr into the system (with molar ratio of $AlBr_3$ still higher than 50%) results in the formation of $[Al_2Br_7]^-$ and $[AlBr_4]^-$ ions in equilibrium with aluminum bromide (in the form of $Al_2Br_6$ and $AlBr_3$). Few of desolvated Br$^-$ exists. Desolvated Na$^+$ and K$^+$ also exist in the system with the amount gradually increasing along with the adding of NaBr/KBr (as shown in Fig. 2b). At this stage, the eutectic molten salt with more than 50 mol% $AlBr_3$ shows Lewis acidity (Fig. 2b) (Supplementary Table 3 for substance species in molten salt). Further increasing the ratio of NaBr/KBr to higher than 50% (the proportion of $AlBr_3$ is less than 50 mol%) (Fig. 2c), even if all $AlBr_3$ forms $[AlBr_4]^-$, there are still large amount of desolvated Br$^-$ from NaBr/KBr, leading to the Lewis basicity in the eutectic molten salt (Supplementary Table 3 for substance species in molten salt). Accordingly, the amount of desolvated Na$^+$ and K$^+$ increase linearly with the adding of NaBr/KBr, while there would be a "suddenly" quick increase of the amount of desolvated Br$^-$ when the ratio of $AlBr_3$ is less than 50 mol%.

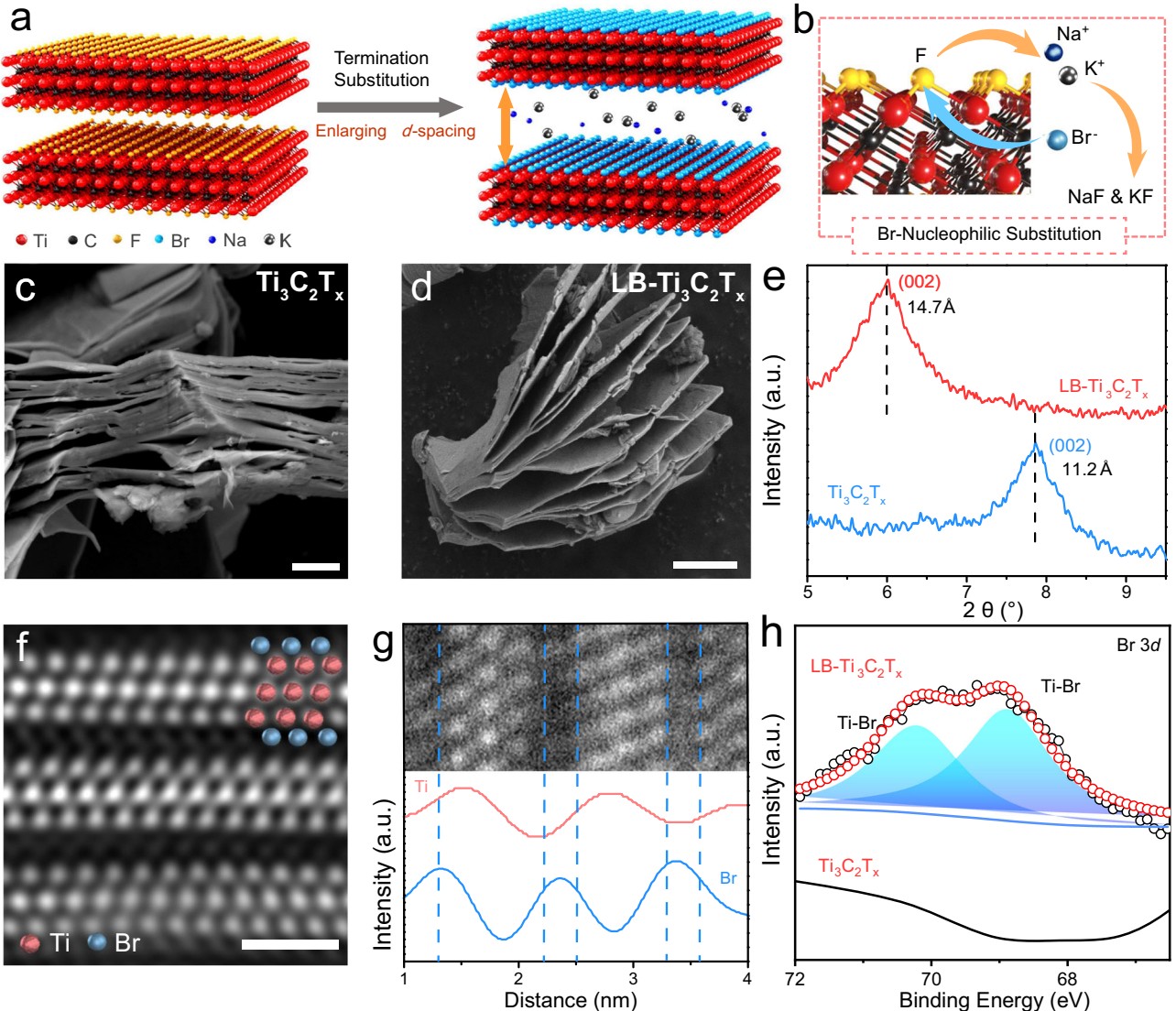

**Fig. 1 | Preparation and characterization of LB-Ti$_3$C$_2$T$_x$. a** Schematic of the preparation of LB-Ti$_3$C$_2$T$_x$. After Lewis-basic halides treatment, Ti$_3$C$_2$T$_x$ is intercalated by desolvated Na$^+$ and K$^+$, inducing the increased interlayer spacing. Simultaneously, the surface -F termination is replaced by desolvated halogen anions. **b** The process of the nucleophilic substitution between desolvated Br$^-$ and F atoms (red balls: titanium, black balls: carbon, yellow balls: fluorine, cyan balls: bromine, blue balls: sodium, silver ball: potassium). **c, d** SEM of multilayer Ti$_3$C$_2$T$_x$ and LB-Ti$_3$C$_2$T$_x$. Scale bar 1 μm. **e** XRD patterns of multilayer Ti$_3$C$_2$T$_x$ and LB-Ti$_3$C$_2$T$_x$. **f** Atomic-resolution high-angle annular dark-field (HAADF) TEM image and the corresponding atomic arrangement. Scale bar 1 nm. **g** Energy dispersive spectrum (EDS) elemental analysis (line scan) of LB-Ti$_3$C$_2$T$_x$. **h** High-resolution Br 3*d* XPS spectra of Ti$_3$C$_2$T$_x$ and LB-Ti$_3$C$_2$T$_x$. After Lewis-basic halides treatment, the peaks of Ti-Br bonds appear and exhibit high intensity. Source data are provided as a Source Data file.

In our experiments, as shown in XRD pattern and SEM images (Fig. 2d and Supplementary Fig. 8), no shift of (002) facet is observed for Ti$_3$C$_2$T$_x$ treated in pure AlBr$_3$ (100 mol% AlBr$_3$), while gradually increased shifts occur for samples treated in 79.9, 60.1 and 48.7 mol% AlBr$_3$. Interestingly, the interlayer spacing enlarges linearly as shown in Fig. 2f. This is in accordance with the variation trend of the amount of desolvated Na$^+$ and K$^+$ in AlBr$_3$/NaBr/KBr molten salts as mentioned previously, demonstrating the increased interlayer spacing of Ti$_3$C$_2$T$_x$ is dominantly attributed to the efficient intercalation of desolvated Na$^+$ and K$^+$ from AlBr$_3$/NaBr/KBr.

The content of -Br termination was analyzed by XPS and EDS. As shown in Fig. 2e, Br element signal is absent for sample treated in 100 mol% AlBr$_3$ as there is no Br$^-$ in molten state AlBr$_3$. For samples treated in 79.9 and 60.1 mol% AlBr$_3$, although Br peaks appear, they are still pretty weak with the Br content calculated to be 0.88 and 1.04 at% for samples treated in 79.9 and 60.1 mol% AlBr$_3$, respectively. Notably, the Br content increases four times (4.18 at%) after treated by 48.7 mol

% AlBr$_3$. Based on the dependence of Br content in Ti$_3$C$_2$T$_x$ on the percentage of AlBr$_3$ in AlBr$_3$/NaBr/KBr molten salts (as shown in Fig. 2f), the variation trend of Br content is simulated to be exponential instead of linear, which is consistent with the tendency of Br$^-$ in AlBr$_3$/NaBr/KBr molten salts. Accordingly, we reasonably propose that the desolvated Br$^-$ in AlBr$_3$/NaBr/KBr molten salts is involved in the termination substitution reaction. This is also supported by theoretical calculation. We estimated the reaction energies (Δ*H*) by the following formulas based on density functional theory (DFT) calculations as:

$$Ti_{27}C_{18}F_{18} + 2NaBr + AlBr_3 \rightarrow Ti_{27}C_{18}F_{17}Br + NaF + NaAlBr_4 \quad (1)$$

$$\Delta H = E(Ti_{27}C_{18}F_{17}Br) + E(NaF) + E(NaAlBr_4) - [E(Ti_{27}C_{18}F_{18}) + 2E(NaBr) + E(AlBr_3)] = -0.336 eV$$

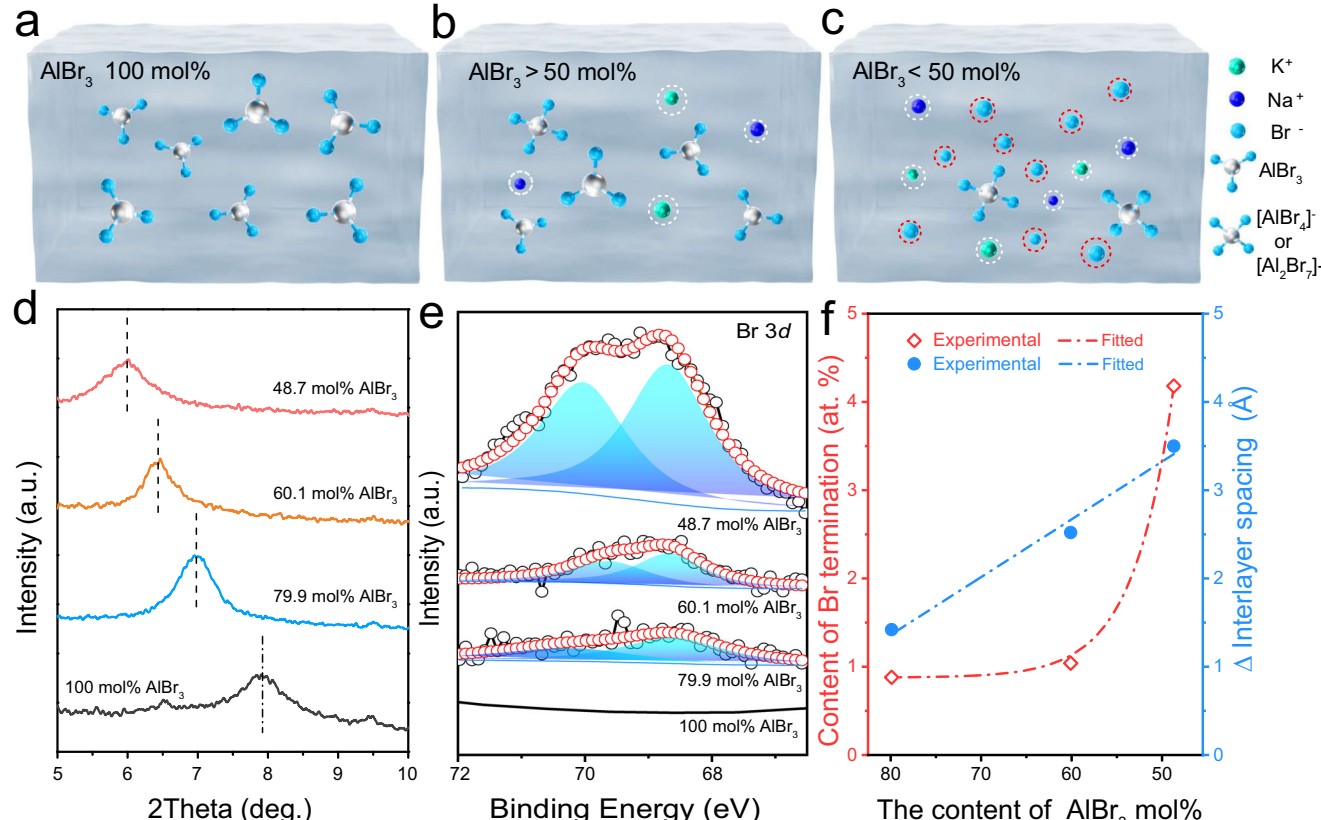

**Fig. 2 | Mechanism of Lewis-basic halides treatment. a–c** Substance component of AlBr$_3$/NaBr/KBr molten salt with different percentages of AlBr$_3$ inside. **a** There is only AlBr$_3$ molecular in pure molten state AlBr$_3$ salt (100 mol% AlBr$_3$). **b** For AlBr$_3$/NaBr/KBr with more than 50 mol% AlBr$_3$, [Al$_2$Br$_7$]$^-$, [AlBr$_4$]$^-$ ions in equilibrium with aluminum bromide (in the form of Al$_2$Br$_6$ and AlBr$_3$) are present with less desolvated Na$^+$ and K$^+$. **c** For AlBr$_3$/NaBr/KBr with less than 50 mol% AlBr$_3$, abundant desolvated Br$^-$, Na$^+$, and K$^+$ exist in molten salts (light green balls: potassium, blue balls: sodium, cyan balls: bromine, silver ball: aluminum). **d, e** XRD patterns and high-resolution Br 3$d$ XPS spectrum of LB-Ti$_3$C$_2$T$_x$ treated by AlBr$_3$/NaBr/KBr molten salts with different component ratios. **f** The variation trend of the content of Br termination and interlayer spacing of LB-Ti$_3$C$_2$T$_x$ after treatment by AlBr$_3$/NaBr/KBr molten salts with different component ratios. The fitted blue curve follows: y$_1$= −0.065x$_1$+6.60. The fitted red curve follows: y$_2$=201028.25*e$^{\left(\frac{-x_2}{7.42}\right)}$+0.88. Source data are provided as a Source Data file.

$$Ti_{27}C_{18}F_{18} + 2KBr + 2AlBr_3 \rightarrow Ti_{27}C_{18}F_{17}Br + KF + KAl_2Br_7 \quad (2)$$

$$\Delta H = E(Ti_{27}C_{18}F_{17}Br) + E(KF) + E(KAl_2Br_7) - [E(Ti_{27}C_{18}F_{18}) + 2E(KBr) + 2E(AlBr_3)] = -0.226eV$$

where $E$ are the total energies of corresponding structures. The calculated $\Delta H$ values are negative, suggesting that the above reaction processes of the Br$^-$ tending to replace F$^-$ ions within Ti$_3$C$_2$F$_2$ are energetically favorable (see details in Methods). In addition, we characterized the unwashed supernatant after AlBr$_3$/NaBr/KBr treatment. According to TEM images (Supplementary Fig. 9), the presence of NaF and KF further demonstrates our assumption on the substitution process between Br and F (Fig. 1b).

For short conclusion, the desolvated ions in AlBr$_3$/NaBr/KBr play an important role on substituting termination and enlarging the interlayer spacing. Only by decreasing the content of AlBr$_3$ to less than 50 mol%, abundant desolvated cations and Br$^-$ exist in the molten salt (this is also demonstrated by the resistance measurement of different molten salts, as shown in Supplementary Fig. 10). Accordingly, Lewis-basic bromide is suggested to utilize on simultaneously tuning the interlayer spacing and termination of Ti$_3$C$_2$T$_x$.

## The universality of Lewis-basic halides treatment on various MXenes

Interestingly, our double-tuning strategy is not only applicable to Ti$_3$C$_2$T$_x$ MXene. For the typical 43 phase MXene-Nb$_4$C$_3$T$_x$, similar regulation of interlayer spacing and surface termination is achieved. As shown in Fig. 3a, Nb$_4$C$_3$T$_x$ prepared by HF etching exhibits a tightly stacked structure with an interlayer spacing of 10.4 Å from XRD pattern (Fig. 3b). After AlBr$_3$/NaBr/KBr treatment, obvious enlarged layered structure of multilayer Nb$_4$C$_3$T$_x$ (LB-Nb$_4$C$_3$T$_x$) is observed (Fig. 3a), with the interlayer spacing increasing to 11.6 Å (Fig. 3b). The presence of Br termination on LB-Nb$_4$C$_3$T$_x$ is validated by XPS (Fig. 3c and Supplementary Figs. 11 and 12, see Supplementary Table 4 for fitting details), as peaks in Br 3$d$ region are split into two Nb-Br bonds. For double metal MXene, typically Mo$_2$Ti$_2$C$_3$T$_x$, the LB-Mo$_2$Ti$_2$C$_3$T$_x$ also performs an enlarged layer structure after treatment (Fig. 3d), in which the interlayer spacing can reach 15.6 Å (5.64°, from XRD pattern in Fig. 3e). Simultaneously, the peak of Br is found in XPS spectra (Supplementary Figs. 13 and 14, see Supplementary Table 5 for fitting details), which could be split into two Mo-Br bonds (Fig. 3f) since the outermost metal layer in Mo$_2$Ti$_2$C$_3$T$_x$ is Mo[38]. As we mentioned, Lewis acid etching is another method for the synthesis of various MXenes[28,39]. However, the as-obtained MXenes usually show a narrow interlayer spacing[22], which is difficult for further delamination[22]. Currently, only TBAOH and $n$-butyllithium are reported for the successful delamination[26,29], while requiring a long-time reaction. Here, by using AlBr$_3$/

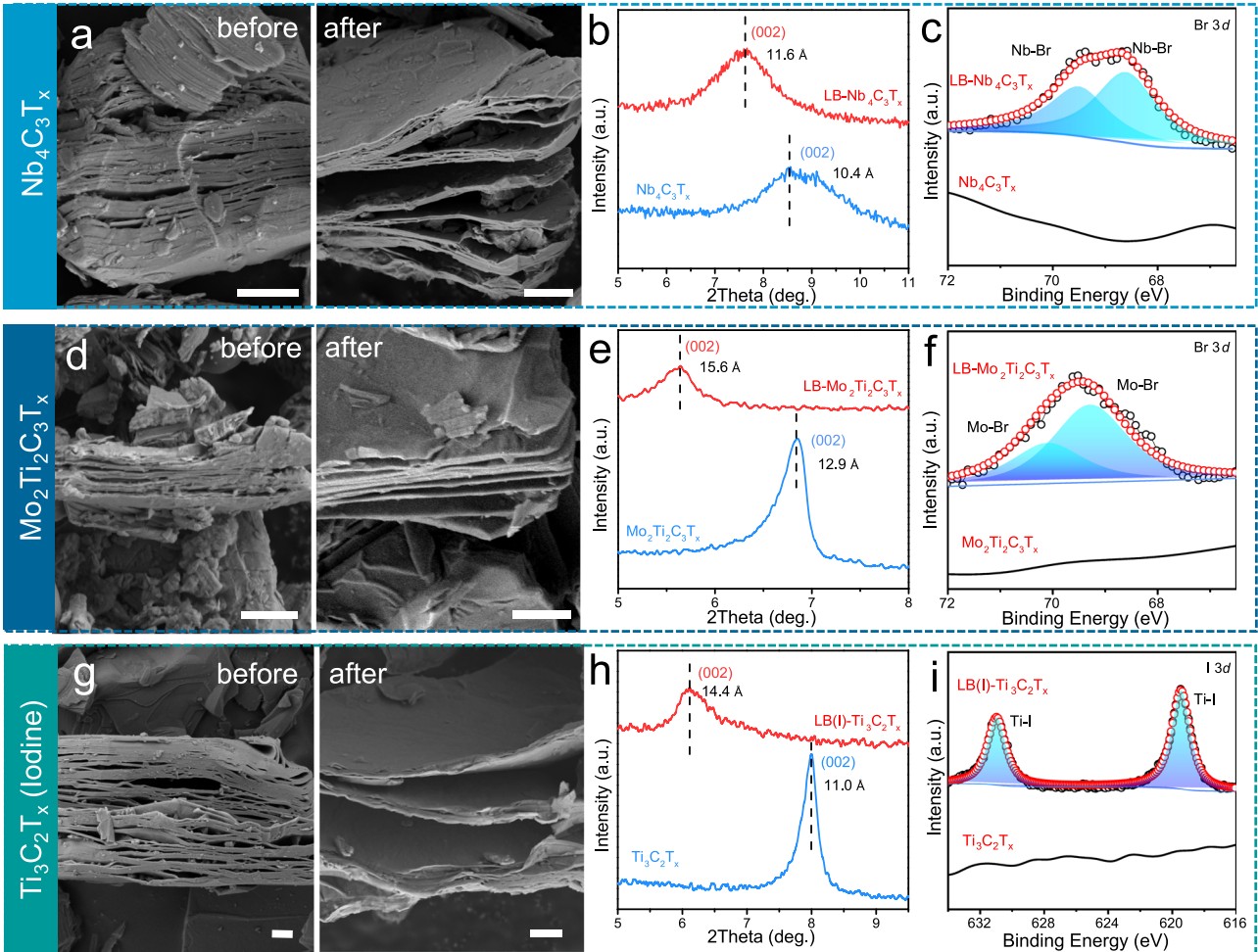

**Fig. 3 | Universality of Lewis-basic halides treatment. a, d, g**, SEM images of $Nb_4C_3T_x$, $Mo_2Ti_2C_3T_x$, and $Ti_3C_2T_x$-Iodine MXenes before and after Lewis-basic halides treatment. Scale bar 1 µm. **b, e, h**, XRD patterns of $Nb_4C_3T_x$, $Mo_2Ti_2C_3T_x$, and $Ti_3C_2T_x$-Iodine MXenes before and after Lewis-basic halides treatment. **c, f, i**, High-resolution XPS spectra of $Nb_4C_3T_x$, $Mo_2Ti_2C_3T_x$, and $Ti_3C_2T_x$-Iodine MXenes. Source data are provided as a Source Data file.

NaBr/KBr treatment, the tightly stacked multilayer MS-$Ti_3C_2T_x$ is apparently opened (Supplementary Fig. 15) along with the introduction of Br termination. We believe our Lewis-basic halides treatment is extremely suitable for Lewis acid etching as the whole etching-intercalation-delamination process could be done in molten salt environment with simply changing the type of molten salts.

Besides Lewis-basic bromides, we also expand our method to Lewis-basic iodines molten salt with the same proportion ($AlI_3$:-NaI:KI = 48.7:15.2:36.1 mol%). It can be seen from the SEM image that the layer structure of $Ti_3C_2T_x$ after treatment is enlarged as well (Fig. 3g), in which the interlayer spacing increase to 14.4 Å (6.13°, from XRD pattern in Fig. 3h). The change in layer spacing is 3.4 Å, almost identical to the change in the Lewis-basic bromides system (3.5 Å). Moreover, XPS demonstrated the existence of Ti-I bond in I $3d$ high-resolution region (Fig. 3i and Supplementary Fig. 16). The above analysis demonstrates the universality of our Lewis-basic halides strategy on simultaneously tuning the interlayer spacing and termination of MXenes.

### Li⁺ storage property of LB-$Ti_3C_2T_x$

To explore the potential application of LB-MXene, multilayer LB-$Ti_3C_2T_x$ (treated by Lewis-basic bromides) is prepared as electrode material for Li⁺ storage. The electrochemical properties are investigated in a 1 M $LiPF_6$+/EC/DMC/EMC electrolyte. During the first lithiation, an irrecoverable reduction peak is observed at

approximately 0.52 V for both $Ti_3C_2T_x$ and LB-$Ti_3C_2T_x$ (Supplementary Fig. 17), corresponding to the solid electrolyte interphase (SEI) formation on the electrode surface[40]. After the first lithiation, the cyclic voltammetry (CV) profiles nearly coincide during the following cycles. At scan rate of 2 mV s⁻¹, the CV profiles of $Ti_3C_2T_x$ and LB-$Ti_3C_2T_x$ are illustrated in Fig. 4a, in which the area of CV curve of LB-$Ti_3C_2T_x$ is significantly larger than that of $Ti_3C_2T_x$, indicating much higher capacity. The specific capacities of $Ti_3C_2T_x$ and LB-$Ti_3C_2T_x$ versus delithiation time and scan rates are shown in Fig. 4b, which are calculated from CV profiles in Supplementary Fig. 18. Specifically, the LB-$Ti_3C_2T_x$ electrode delivers a specific capacity of 168 mAh g⁻¹ at a scan rate of 0.5 mV s⁻¹, which corresponds to 215 F g⁻¹ (Supplementary Table 6). At the same scan rate, the $Ti_3C_2T_x$ electrode achieves a much lower capacity of 78 mAh g⁻¹ (99 F g⁻¹). As the scan rate increases to 100 mV s⁻¹, the $Ti_3C_2T_x$ electrode shows a specific capacity of 14 mAh g⁻¹ (Supplementary Table 7), while the specific capacity of LB-$Ti_3C_2T_x$ is 42 mAh g⁻¹, which is three times the capacity of $Ti_3C_2T_x$, demonstrating the high rate capability of LB-$Ti_3C_2T_x$. The voltage profiles of LB-$Ti_3C_2T_x$ electrode from galvanostatic tests are shown in Fig. 4c. A high Coulombic efficiency (96%) is achieved and a maximum capacity of 229 mAh g⁻¹ is recorded at a specific current of 0.1 A g⁻¹, which is higher than the reported value for Br-terminated $Ti_3C_2T_x$[41].

We believe the high capacity and rate capability could be attributed to the enlarged interlayer spacing that provides fast and more Li⁺

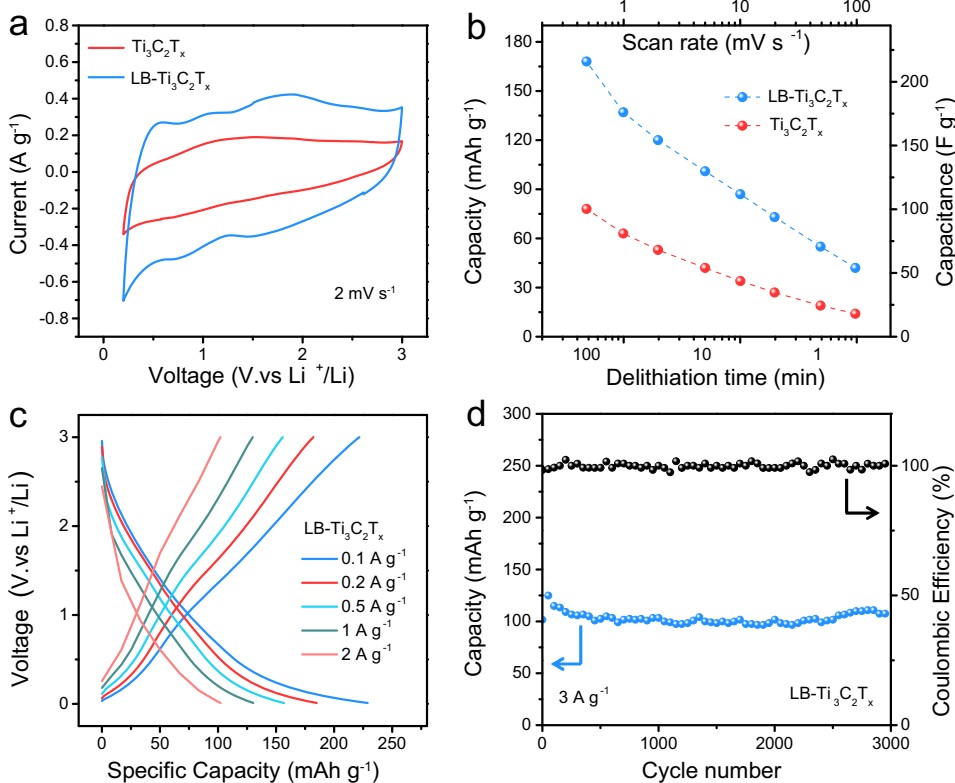

**Fig. 4 | Electrochemical energy storage properties of LB-Ti$_3$C$_2$T$_x$. a** CV profiles of Ti$_3$C$_2$T$_x$ and LB-Ti$_3$C$_2$T$_x$ at scan rate of 2 mV s$^{-1}$. **b** The comparison of capacity of Ti$_3$C$_2$T$_x$ and LB-Ti$_3$C$_2$T$_x$ at various scan rates and delithiation time. **c** Voltage profiles of LB-Ti$_3$C$_2$T$_x$ at various current densities. **d** Cycling stability of LB-Ti$_3$C$_2$T$_x$ at 3 A g$^{-1}$. Source data are provided as a Source Data file.

transport. This can be proved by electrochemical impedance spectroscopy (EIS, Supplementary Fig. 19b), which shows a charge transfer resistance (R$_{ct}$) of 80.2 Ω of LB-Ti$_3$C$_2$T$_x$ compared to 277.1 Ω of Ti$_3$C$_2$T$_x$. In addition, the substitution of F termination by Br is beneficial to good Li$^+$ storage performance as F largely hinders the Li$^+$ transport[15,41]. The cycling stability of LB-Ti$_3$C$_2$T$_x$ at high current densities of 3 A g$^{-1}$ is shown in Fig. 4d. After 3000 cycles, it still shows a specific capacity of 120 mAh g$^{-1}$ with a capacity retention over 95%.

In conclusion, we present a general method that is capable of simultaneously tuning the interlayer spacing and termination of MXenes by Lewis-basic halides. The desolvated halogen ions are involved in the termination substitution via nucleophilic reaction. On the other hand, the synergistic effect of termination substitution and cations (abundant desolvated Na$^+$ and K$^+$) intercalation are responsible for the enlarged interlayer spacing. This method is applicable to various MXenes, including 32/43 phase MXenes, mono/double MXenes, and MXenes synthesized by HF or Lewis acid etching. The LB-Ti$_3$C$_2$T$_x$ is demonstrated as the electrode material for Li$^+$ storage, which shows a high specific capacity of 229 mAh g$^{-1}$ at 0.1 A g$^{-1}$ (almost two times higher than pristine Ti$_3$C$_2$T$_x$) with high rate capability and long-term stability. Our method provides a strategy on engineering the interface and surface chemistry of MXenes, which may have impact on other properties. 2D materials beyond MXenes may be also regulated by this method, which deserves further study.

## Methods
### Synthesis of Ti$_3$AlC$_2$ MAX
TiC (99%, Macklin), Al (99%, Tianjin Damao Chemical Reagent Co., Ltd) and Ti (99%, Macklin) (mass ratio = 2:1.3:1) powders were mixed and grounded with zirconia ball at 70 rpm for 18 h. The mixture was annealed at 1400 °C for 2 h under the atmosphere of argon, with the heating/cooling rate of 3 °C min$^{-1}$. In general, 1 g of mixture powder

was washed by 10 ml HCl (38 wt.%, Sigma-Aldrich) to remove the redundant aluminum. Then the Ti$_3$AlC$_2$ MAX powder (400 mesh) was obtained by vacuum filtration and dried under vacuum at 60 °C for 12 h.

### Synthesis of multilayer MXenes
Multilayer MXenes were prepared by selectively etching A element from MAX. Ti$_3$C$_2$T$_x$: 1 g of Ti$_3$AlC$_2$ powder was added to 18 mL HCl, 3 mL HF (49 wt.%, Sigma-Aldrich) and 9 mL deionized (DI) water, followed by stirring for 24 h at 35 °C at 400 rpm. Mo$_2$Ti$_2$C$_3$T$_x$: 1 g of Mo$_2$Ti$_2$AlC$_3$ powder (11 Technology Co., Ltd) was added to 10 mL HF and stirred at 55 °C for 96 h. Nb$_4$C$_3$T$_x$: 1 g of Nb$_4$AlC$_3$ powder (11 Technology Co., Ltd) was added to 10 mL HF and stirred at room temperature (RT) for 96 h. The different multilayer MXenes were obtained by washing the etched sediment with DI water for 5-6 times, until the pH of the supernatant reached 6. (Molten salt) MS-Ti$_3$C$_2$T$_x$: 2.1 g of CuCl$_2$ powder (99.99%, Macklin) and 1 g of Ti$_3$AlC$_2$ powder were mixed and grounded for 10 min. Then 0.76 g of KCl (≥99.5% Sinopharm Chemical Reagent Co., Ltd) and 0.6 g of NaCl (≥99.5% Sinopharm Chemical Reagent Co., Ltd) were added and grounded for 10 min. The mixture was heated at 700 °C for 24 h in an alumina tube under argon with a heating rate of 5 °C min$^{-1}$. Multilayer MS-Ti$_3$C$_2$T$_x$ was obtained by washing the sediment with DI water and 0.1 M ammonium persulfate (≥98%, Aladdin) solution. All MXenes powders were obtained by vacuum drying at 60 °C for 12 h.

### Synthesis of LB-Ti$_3$C$_2$T$_x$ and other MXenes
All experiments were carried out in argon atmosphere. 150 mg Ti$_3$C$_2$T$_x$ powder was dispersed in the molten salt (about 9.70 g) with a composition of AlBr$_3$ (98%, Macklin): NaBr (99.9%, Macklin): KBr (SP, Macklin) = 48.7:15.2:36.1 mol%, followed by stirring for 6 h at 400 rpm at 230 °C. After cooling, the multilayer LB-Ti$_3$C$_2$T$_x$ (Lewis-basic halides

treated $Ti_3C_2T_x$) was obtained by washing the sediment with anhydrous diethyl ether (≥99.5%, Chengdu Chron Chemical Co., Ltd), tetrahydrofuran (≥99.5%, Shanghai Titan Scientific Co., Ltd) and DI water for three times, respectively. Then, the $Ti_3C_2T_x$ was treated by eutectic molten salt $AlBr_3/NaBr/KBr$ with different molar ratio ($AlBr_3$:NaBr:KBr = 79.9:5.9:14.2 mol%, 60.1:11.7:28.2 mol%, 40.1:17.6:42.3 mol%) in the similar process, respectively. It is noticed that there is some undissolved salt when the $AlBr_3$ ratio is 40.1 mol% (Supplementary Fig. 20). LB (iodine)-$Ti_3C_2T_x$ was operated in the proportion ($AlI_3$ (99.99%, Aladdin): NaI (99.5%, Macklin): KI (SP, Macklin) = 48.7:15.2:36.1 mol%), followed by stirring for 6 h at 400 rpm at 230 °C. The monolayer LB-$Ti_3C_2T_x$ can be obtained by sonication (<10 °C to avoid oxidation) for 1 h. Different types of LB-MXenes were operated in the same way as LB-$Ti_3C_2T_x$ MXene.

## Computational method

All density functional theory (DFT) calculations were performed by using the projector augmented wave (PAW) method[42], as implemented in the Vienna ab initio Simulation Package (VASP)[43,44]. Generalized gradient approximation (GGA) parameterized by Perdew-Burke-Ernzerhof (PBE) form[45] was employed for evaluating the electron exchange correlation energy. The DFT-D3 method of Grimme was used to describe the weak dispersion forces[46]. The energy cutoff for the plane waves was set to 500 eV. The Monkhorst–Pack scheme[47] with $3 \times 3 \times 1$ $k$ point meshes were used for the sampling in the irreducible Brillouin zone. The structural parameters and all the atoms were fully optimized until the Hellman-Feynman forces were less than 0.01 eV/Å. The bare $Ti_3C_2$ monolayer was built by removing Al layers from bulk phase of $Ti_3AlC_2$ ($a = b = 9.333$ Å, $c = 18.699$ Å, $\alpha = \beta = 90°$, $\gamma = 120°$, obtained from the structure of #153266 in ICSD database). A vacuum layer thickness of 25 Å was applied to all the slab models. Based on the optimized $Ti_3C_2$ monolayer, the $Ti_3C_2F_2$ ($a = b = 9.179$ Å; $c = 32.253$ Å; $\alpha = \beta = \gamma = 90°$) monosheet was further built.

## Material characterization

The crystal architecture was identified by X-ray diffraction (XRD) patterns (D/max 2600). In order to characterize the samples morphology, scanning electron microscopy (SEM, JSM-7600F) with an energy dispersive spectrum (EDS) was utilized. Transmission electron microscope (TEM) images were obtained from Tecnai G2 F20 S-TWIN, FEI (200 kV). XPS was conducted using PHI VersaProbe 5000 instrument (Physical Electronics) with a 100 μm and 25 W monochromatic Al-Kα (1486.6 eV) X-ray source. The local element distribution (line scans) was analyzed by highly efficient energy dispersive spectrum (EDS) spectroscopy at 300 kV with a point-to-point resolution of 0.2 nm and a maximum resolution of 0.06 nm in a high-angle annular dark-field (HAADF) high-resolution scanning transmission electron microscopy (STEM). The samples for cross-sectional transmission electron microscopy (TEM) were prepared by FEI HELIOS NanoLab 600i Focused Ion Beam (FIB) system.

## Electrochemical measurements

The electrodes consisted of anode material, acetylene black and N-methyl-2-pyrrolidone (NMP) solution with 5% PVDF in the mass ratio of 7:1.5:1.5. Then use a squeegee to evenly coat the paste prepared above the copper foil. The copper foil was dried in vacuum at 60 °C for 12 h. Finally, the working electrode was made by cutting copper foil into a 13 mm diameter disc and lithium sheet was used as a counter electrode. The loading mass of the active substance is about 0.7 mg cm$^{-2}$. The electrolyte was a mixture of 1 M $LiPF_6$, ethylene carbonate (EC), methyl carbonate (DMC) and ethyl carbonate (EMC) solutions (EC/DMC/EMC ratio of 1:1:1 by volume). The batteries were assembled in an argon-filled glove box with oxygen and water content less than 0.1 ppm. The electrodes were assembled into CR2016 coin-type cells for electrical performance testing on an automatic battery tester (LAN-

2100) at RT. CV profiles were obtained in an electrochemical workstation (Bioologic, France) and EIS was conducted with a frequency range of 200 kHz–10 mHz and an AC perturbation amplitude of 5 mV.

The specific capacitance and capacity of the CV profiles are calculated. The formulas are as following:

$$C = \frac{\int_0^{V/s} |i| dt}{m} \tag{3}$$

$$Q = CV \tag{4}$$

$$Q_m = \frac{Q_c}{3.6} \tag{5}$$

where $C$ is the specific capacitance (F g$^{-1}$), $i$ (A) is the current changed by time $t$ (s), $m$ is the active material mass (g), $V$ is the voltage window (V), $s$ is the scan rate (V s$^{-1}$), $m$ is the active material mass (g), $Q_c$ (C g$^{-1}$) and $Q_m$ (mAh g$^{-1}$) are the specific capacities.

## Reporting summary

Further information on research design is available in the Nature Portfolio Reporting Summary linked to this article.

## Data availability

Source Data are provided with this paper.

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

## Acknowledgements

This work is supported by the Outstanding Scholarship Foundation of UESTC (A1098531023601243) (X.X.), and the Sichuan Science and Technology Support Program (2021JDTD0026) (X.X.).

## Author contributions

X.X. conceived the idea and supervised the project. T.Z. carried out the experiments. X.Z. conducted the theoretical calculation under the supervision of L.Z. T.Z., L.C., H.W., N.L., and X.X. analyzed the data. T.Z., L.C., X.Z., H.W., N.L., L.Z., and X.X. wrote the manuscript. All of the authors commented on the manuscript.

## Competing interests

The authors declare no competing interests.
