## [Peer Review File · Nature Communications]

Simultaneously tuning interlayer spacing and termination of MXenes by Lewis-basic halidesREVIEWER COMMENTS

Reviewer #1 (Remarks to the Author):

The authors reported a new and effective strategy to regulating the terminations and interlayer spacing of MXenes simultaneously by Lewis-basic halides. This strategy can easily change terminations from F to Br and I with enlarged interlayer spacing at a relatively low temperature, which cannot be achieved in previous researches. The enlarged interlayer spacing and regulated termination species for MXenes are responsible for improved electrochemical performance. And I believe that this strategy can be broadened to more MXenes and applicable to other fields. Therefore, I recommend publication of this work after a minor revision.

1. Why the enlarged interlayer spacing of various MXenes (mono/double metal MXenes, 32 and 43 phase MXenes) treated by Lewis-basic halides are different?
2. b value is an important factor for energy storage. Therefore, the b value of LB-Ti₃C₂T_x needs to be calculated to determine whether it is a battery/capacitive behavior.
3. Can the authors comment on the industrial prospects of this Lewis-basic halides method? How's the yield and can it be scale-up?

Reviewer #2 (Remarks to the Author):

The authors present a general method that is capable of simultaneously tuning the interlayer spacing and termination of MXenes by Lewis-basic halides. They claim that the abundant desolvated Na⁺ and K⁺ in Lewis-basic halides are responsible for the highly enlarged interlayer spacing through the efficient intercalation process, the desolvated halogen ions are involved in the termination substitution via nucleophilic reaction. This method paves the way for engineering the interfacial and surface chemistry of MXenes. However, there are some issues should be considered.

- 1) It is well known that Mxene will self-oxidize at high temperature even in argon and it is relatively stable at low temperature. Through the treatment of eutectic molten salt, LB-Ti₃C₂T_x MXene shows more titanium dioxide species than pristine Ti₃C₂T_x in Fig. S6. Does this mean that this method has little effect on inhibiting MXene self-oxidation?
- 2) The (002) facet and the content of -Br termination is gradually increased shifts occur for samples treated in 79.9, 60.1 and 48.7 mol% AlBr₃ in Fig 2f. What will be the of (002) facets and the content -br terminations in samples treated with eutectic molten salt containing less AlBr₃?
- 3) Is there any change of interlayer spacing for LB-Ti₃C₂T_x in the process of lithium ion insertion and removal? Will sodium ions and potassium ions migrate from the interlayer?
- 4) We all know that Li⁺ is much smaller than Na⁺ and K⁺. Perhaps the storage of Na⁺ or K⁺ can better show the advantages of LB-Ti₃C₂T_x materials.

Reviewer #3 (Remarks to the Author):

This is paper that could possibly be published. But more information is needed (especially XPS) to shore up their conclusions. I would like the authors to address some of the following questions/comments.

The authors are emphasizing the uniform nature of their terminations, but do not show the XPS of the O region. In the Ti-region they fit for O and OH. It is imperative for them to show the O region in XPS as well and discuss it. The authors should show ALL XPS of ALL elements and discuss them.

The authors are strongly emphasizing the expanded interlayer space but attribute the increase in d to the wrong reasons. Higher, d-spacings have been achieved for water etched MXenes as well. So there is nothing special here.

The authors claim: " If the above two 3 processes occur in aqueous solution, the overall activation energy increases due to the energy consumption for the necessary step of ion de-solvation, which limits the reaction rate according to the Arrhenius equation" This is simply not true. In aqueous solutions, solvated ions readily intercalate at room temperature. That's how one obtains 16 Å spacing.

The authors claim, " In this context, the molten salt system is of particular interest because "naked" ions could be directly involved in reaction, which is beneficial for largely increasing interlayer spacing and efficient termination substitution." I don't get logic here. If higher interlayer spacings are beneficial, then ideally one should prefer solvated ions over naked ions as the former have larger diameters than the latter.

The authors claim, "Such a large interlayer spacing is rarely achievable via the intercalation of inorganic cations, as interlayer spacing ranging between 12.3 Å and 13.6 Å is generally reported for Li⁺ intercalated multilayer Ti₃C₂T_x" This increase in d is mostly because of -Br or -I terminations and not the cations per se." Here again it appears the authors are really not familiar with the literature on interlayer spacings in MXenes. In the LiF/HCl method 16 Å is routine.

Along the same lines the hype here is unwarranted. The authors claim "Although numerous efforts have been devoted to the regulation of Ti₃C₂T_x MXene, finding a general and facile route to tuning interlayer spacing and surface termination of various MXenes remains challenging while demanded." And then proceed to describe a process where the MAX phase needs first to be etched with HF, then placed in one molten bath, taken out of it, washing the powders and then placed in a second molten bath, before taking it out and washing it, etc. Whatever you call this, it is not facile by any stretch of the imagination. Their dismissal of previous work is also off base. If I compare their method to say DMSO all carried out in water at RT, there is no doubt which is the more facile. Similarly, dismissing previous molten salt as being too high a temperature is also off base. If I look at Table S1, there is serious reduction in the Ti-C Mxene signal. That is worrisome. DO the authors have an explanation?

Why are the authors not showing the Nb results in Table S3?

Why did the authors use tetrahydrofuran and diethyl ether for washing?

In short, this paper needs significant revision before being accepted.

Response Notes

(Simultaneously tuning interlayer spacing and termination of MXenes by Lewis-basic halides)

We appreciate for the editor and reviewers' constructive suggestions to our manuscript. All the suggestions are very helpful for us to improve our research. We have addressed all comments of the reviewers, and our responses to the reviewer's comments are listed below point-by-point. The revised sections have been highlighted in the revised manuscript.

Reviewer #1

The authors reported a new and effective strategy to regulating the terminations and interlayer spacing of MXenes simultaneously by Lewis-basic halides. This strategy can easily change terminations from F to Br and I with enlarged interlayer spacing at a relatively low temperature, which cannot be achieved in previous researches. The enlarged interlayer spacing and regulated termination species for MXenes are responsible for improved electrochemical performance. And I believe that this strategy can be broadened to more MXenes and applicable to other fields. Therefore, I recommend publication of this work after a minor revision.

Response: We appreciate the efforts the reviewer has spent on our manuscript and thanks for the reviewer's positive comments. The point-by-point response has been done as follows.

Comment 1. *Why the enlarged interlayer spacing of various MXenes (mono/double metal MXenes, 32 and 43 phase MXenes) treated by Lewis-basic halides are different?*

Response: According to our experimental results, the increase in interlayer spacing of $\text{Ti}_3\text{C}_2\text{T}_x$, $\text{Nb}_4\text{C}_3\text{T}_x$ and $\text{Mo}_2\text{Ti}_2\text{C}_3\text{T}_x$ is 3.4, 1.2 and 2.7 Å, respectively. In general, the intercalation process is largely controlled by terminations, as it affects the surface charges. We infer that the difference of interlayer spacing of the above MXenes is attributed to the different surface negative charges (*Chem. Mater.* 2022, 34, 678-693).

Comment 2. *b value is an important factor for energy storage. Therefore, the b value of LB- $\text{Ti}_3\text{C}_2\text{T}_x$ needs to be calculated to determine whether it is a battery/capacitive*

behavior.

Response: Thanks for the reviewer's suggestion. We have calculated b value in the revised manuscript. According to the power-law relationship $i=av^b$ (*Nat. Mat.* 2013, 12, 518-522), the correlation between the peak current (i) and the sweep rate (v) can be studied to distinguish the storage process of the charge, where a and b denote arbitrary coefficients. When b is close to 0.5, it shows diffusion-controlled charge storage. When it is close to 1.0, it shows capacitance-controlled charge storage. In our work, the calculated b value (0.87) is close to 1 (Fig. R1-1), indicating that the LB-Ti₃C₂T_x electrode is dominated by capacitance-controlled charge storage.

Fig. R1-1. Change of the peak current with the potential scan rate in log scale. (Fig. S21 in revised manuscript)

Action: We have added the Fig. S21 and accompanied with the analysis in Supplementary Information:

“According to the power-law relationship $i=av^b$, the correlation between the peak current (i) and the sweep rate (v) can be studied to distinguish the storage process of the charge, where a and b denote arbitrary coefficients¹. When b is close to 0.5, it shows diffusion-controlled charge storage. When b is close to 1.0, it shows capacitance-controlled charge storage. In our work, the calculated b value (0.87) is close to 1, indicating that the LB-Ti₃C₂T_x electrode is dominated by capacitance-controlled charge storage.”

Reference

1. Augustyn, V. et al. High-rate electrochemical energy storage through Li^+ intercalation pseudocapacitance. *Nat. Mater.* 12, 518–522 (2013).

Comment 3. *Can the authors comment on the industrial prospects of this Lewis-basic halides method? How's the yield and can it be scale-up?*

Response: At present, the industry of molten salt synthesis is pretty mature. For instance, the aluminum and rare earth metals are mainly prepared by molten salt electrolysis (*Rare Metals* 2016, 35, 811-825). We believe our Lewis-basic halides method has high potential for commercialization through borrowing the current industrial experience of molten salt synthesis.

The yield of this method is about 92%. And it could be scaled up by using large volume stirrer.

Reviewer #2

The authors present a general method that is capable of simultaneously tuning the interlayer spacing and termination of MXenes by Lewis-basic halides. They claim that the abundant desolvated Na^+ and K^+ in Lewis-basic halides are responsible for the highly enlarged interlayer spacing through the efficient intercalation process, the desolvated halogen ions are involved in the termination substitution via nucleophilic reaction. This method paves the way for engineering the interfacial and surface chemistry of MXenes. However, there are some issues should be considered.

Response: We appreciate the efforts the reviewer has spent on our manuscript and thanks for the reviewer's positive comments. The point-by-point response has been done as follows.

Comment 1. *It is well known that MXene will self-oxidize at high temperature even in argon and it is relatively stable at low temperature. Through the treatment of eutectic molten salt, LB- $\text{Ti}_3\text{C}_2\text{T}_x$ MXene shows more titanium dioxide species than pristine $\text{Ti}_3\text{C}_2\text{T}_x$ in Fig. S6. Does this mean that this method has little effect on inhibiting MXene self-oxidation?*

Response: According to XPS, the proportion of TiO_2 increased a bit from 9.34% to 13.24% (Table R2-1). And there is no peak of TiO_2 in the XRD pattern (Fig. R2-1),

indicating that the oxidation degree of LB-Ti₃C₂T_x is still relatively low. Although there is supposed to be no water/oxygen-containing molecules in molten salt, the O and OH terminations may self-oxidize MXene, which should be the reason for the small increase of TiO₂ in MXene after treatment. Based on the above analysis, although only a low degree of oxidation was detected, we think our method couldn't completely inhibit MXene self-oxidation.

Table R2-1. XPS fitting results of Ti₃C₂T_x and LB-Ti₃C₂T_x. (Table S2 in revised manuscript)

Ti ₃ C ₂ T _x	BE (eV)	FWHM (eV)	Fraction	Assigned to
Ti 2p _{3/2} (Ti 2p _{1/2})	455.0 (461.0)	1.20 (1.20)	29.48	(O/OH)-Ti-C
	455.8 (461.6)	1.50 (1.50)	38.42	(O/OH)-Ti ⁺² -C
	456.9 (462.5)	1.32 (1.32)	17.61	(O/OH)-Ti ⁺³ -C
	458.8 (463.6)	1.70 (1.70)	9.34	TiO ₂
	460.3 (464.2)	1.10 (1.10)	5.15	Ti-F
C 1s	281.8	0.71	28.50	Ti-C
	284.8	1.43	48.56	C-C
	285.3	1.05	16.76	CH _x
	286.2	1.16	5.59	C-O
	289.1	1.00	0.59	-COO
O 1s	529.7	1.01	27.88	TiO ₂
	530.7	1.20	18.43	Ti-O _x
	531.7	1.25	12.76	Ti-OH
	532.6	1.09	6.32	C-O
	534.0	2.74	34.61	H ₂ O

LB-Ti ₃ C ₂ T _x	BE (eV)	FWHM (eV)	Fraction	Assigned to
Ti 2p _{3/2} (Ti 2p _{1/2})	454.9 (461.1)	1.00 (1.00)	20.73	(O/OH and Br)-Ti-C
	455.7 (461.7)	1.40 (1.40)	40.58	(O/OH and Br)-Ti ⁺² -C
	456.9 (462.7)	1.60 (1.60)	24.07	(O/OH and Br)-Ti ⁺³ -C
	459.2 (464.3)	1.60 (1.60)	13.24	TiO ₂
	460.4 (465.2)	1.00 (1.00)	1.38	Ti-F
C 1s	281.9	1.00	10.10	Ti-C
	284.8	1.20	51.44	C-C
	285.2	1.00	22.41	CH _x
	285.9	1.40	14.30	C-O
	289.2	0.92	1.75	-COO
O 1s	529.8	0.80	9.65	TiO ₂
	530.8	1.17	20.71	Ti-O _x
	531.8	1.17	17.09	Ti-OH
	532.7	1.34	39.27	C-O
	533.6	1.11	13.28	H ₂ O
Br 3d	68.9 (70.2)	1.56 (1.56)	1.00	Ti-Br

BE: Binding Energy

Fig. R2-1. XRD patterns of TiO₂, Ti₃C₂T_x, and LB-Ti₃C₂T_x.

Comment 2. The (002) facet and the content of -Br termination is gradually increased shifts occur for samples treated in 79.9, 60.1 and 48.7 mol% AlBr_3 in Fig 2f. What will be the of (002) facets and the content -Br terminations in samples treated with eutectic molten salt containing less AlBr_3 ?

Response: We thank the reviewer for this constructive suggestion. As shown in Fig. R2-2a, when the added AlBr_3 is less than 48.7 mol% (for instance, 40.1 mol%), sediments appear at the bottom of the solution (Fig. R2-2a), indicating the solubility limit (the content of AlBr_3 in NaBr/KBr/AlBr_3) may be around 48.7 mol%. In this context, treating $\text{Ti}_3\text{C}_2\text{T}_x$ in this molten salts (40.1 mol% ArBr_3 with sediments) cannot further increase the interlayer spacing and the content of -Br terminations as shown in Fig. R2-2b and 2c. This is due to the actual content of Na^+ , K^+ , and Br^- in 40.1 mol% ArBr_3 is not as high as in 48.7 mol% ArBr_3 .

Fig. R2-2. **a**, XRD patterns of $\text{Ti}_3\text{C}_2\text{T}_x$ and (40.1 mol% AlBr_3 and 48.7 mol% AlBr_3) LB- $\text{Ti}_3\text{C}_2\text{T}_x$. **b**, Photograph of the $\text{AlBr}_3/\text{NaBr/KBr}$ molten salt with 40.1 mol% of AlBr_3 . Sediments are clearly seen at the bottom. **c**, EDS analysis of LB- $\text{Ti}_3\text{C}_2\text{T}_x$ (40.1 mol% AlBr_3).

Action: In the Supplementary Information, we have made an explanation in Fig. S21:

“When the added AlBr_3 is less than 48.7 mol% (for instance, 40.1 mol%), sediments appear at the bottom of the solution (Fig. S20), indicating the solubility limit (the content of AlBr_3 in NaBr/KBr/AlBr_3) may be around 48.7 mol%.”

Fig. S20. Photograph of the $\text{AlBr}_3/\text{NaBr}/\text{KBr}$ molten salt with 40.1 mol% of AlBr_3 .

Comment 3. *Is there any change of interlayer spacing for $\text{LB-Ti}_3\text{C}_2\text{T}_x$ in the process of lithium ion insertion and removal? Will sodium ions and potassium ions migrate from the interlayer?*

Response: We thank the reviewer for pointing this out. The interlayer spacing decreased from 14.4 (6.13°) to 13.8 Å (6.38°) during lithium deintercalation, and increases from 13.8 (6.38°) back to 14.3 Å (6.17°) during lithium intercalation (Fig. R2-3), which means the interlayer structure of $\text{LB-Ti}_3\text{C}_2\text{T}_x$ is pretty stable during cycling.

Further, we test the EDS of $\text{LB-Ti}_3\text{C}_2\text{T}_x$ after cycling. As the charging and discharging cycles increase, the atomic content of Na^+ and K^+ decreased. (Table R2-2). Therefore, we speculate that Na^+ and K^+ will migrate (replaced by Li^+) from the interlayer during cycling.

Fig. R2-3. Ex-situ XRD patterns of $\text{LB-Ti}_3\text{C}_2\text{T}_x$ electrode at different potential during the charge and discharge processes. (Fig. S22 in revised manuscript)

Table R2-2. EDS element content of LB-Ti₃C₂T_x and LB-Ti₃C₂T_x after cycling.

Element (at. %)	Pristine LB-Ti ₃ C ₂ T _x	LB- Ti ₃ C ₂ T _x after cycling
Ti	96.67	99.23
Na	2.90	0.57
K	0.43	0.20

Action: In the Supplementary Information, we have added Fig. S22 and made an explanation:

“The interlayer spacing decreased from 14.4 (6.13°) to 13.8 Å (6.38°) during lithium deintercalation, and increases from 13.8 (6.38°) back to 14.3 Å (6.17°) during lithium intercalation (Fig. S22), which means the interlayer structure of LB-Ti₃C₂T_x is pretty stable during cycling.”

Comment 4. *We all know that Li⁺ is much smaller than Na⁺ and K⁺. Perhaps the storage of Na⁺ or K⁺ can better show the advantages of LB-Ti₃C₂T_x materials.*

Response: Based on the reviewer’s suggestion, we also tested the performance of Na ion battery of LB-Ti₃C₂T_x. At scan rate of 1 mV s⁻¹, the CV profiles of Ti₃C₂T_x and LB-Ti₃C₂T_x are illustrated in Fig. R2-4a, in which the area of CV curve of LB-Ti₃C₂T_x is 1.6 times larger than that of Ti₃C₂T_x, indicating higher capacity. The voltage profiles of Ti₃C₂T_x electrode from galvanostatic tests are shown in Fig. R2-4b. A maximum capacity of 55 mAh g⁻¹ is recorded for pristine Ti₃C₂T_x at a specific current of 0.1 A g⁻¹ in Na⁺ battery. In contrast, the LB-Ti₃C₂T_x electrode achieves a much higher capacity of 122 mAh g⁻¹ (Fig. R2-4c). The above results show that LB-Ti₃C₂T_x electrode has better storage of Na⁺ than Ti₃C₂T_x, which is similar to the enhanced Li⁺ storage behavior in LB-Ti₃C₂T_x.

Fig. R2-4. a, The CV profiles of $\text{Ti}_3\text{C}_2\text{T}_x$ and $\text{LB-Ti}_3\text{C}_2\text{T}_x$ at scan rate of 1 mV s^{-1} . Voltage profiles of (b) $\text{Ti}_3\text{C}_2\text{T}_x$ and (c) $\text{LB-Ti}_3\text{C}_2\text{T}_x$ at various current densities.

Reviewer #3

This is paper that could possibly be published. But more information is needed (especially XPS) to shore up their conclusions. I would like the authors to address some of the following questions/comments.

Response: We appreciate the efforts the reviewer has spent on our manuscript. The point-by-point response has been done as follows.

Comment 1. *The authors are emphasizing the uniform nature of their terminations, but do not show the XPS of the O region. In the Ti-region they fit for O and OH. It is imperative for them to show the O region in XPS as well and discuss it. The authors should show all XPS of all elements and discuss them.*

Response: According to the reviewer's suggestion, we have added the O 1s regions of $\text{Ti}_3\text{C}_2\text{T}_x$ and $\text{LB-Ti}_3\text{C}_2\text{T}_x$, the XPS spectra of all elements and the corresponding analyses in revised Supplementary Information. The O 1s region (Fig. R3-1c and 1f) could be assigned to five peaks corresponding to TiO_2 , Ti-O_x , Ti-OH , C-O and H_2O , respectively. After Lewis basic halides treatment, the peaks of Ti-O_x and Ti-OH retained, indicating that O termination wasn't substituted by -Br. Compared with pristine $\text{Ti}_3\text{C}_2\text{T}_x$, the strength of C-O in $\text{LB-Ti}_3\text{C}_2\text{T}_x$ increased, as shown in Fig. R3-1f. Considering anhydrous diethyl ether and tetrahydrofuran which contain C-C and C-O bonds are used for removing salts in our experiment, it is reasonable to speculate that there is a small amount of organic solvent residue on the surface of $\text{LB-Ti}_3\text{C}_2\text{T}_x$ in the process of washing salt.

Nb₄C₃T_x and LB-Nb₄C₃T_x

In the high resolution XPS spectra of Nb₄C₃T_x and LB-Nb₄C₃T_x, the Nb 3d region (Fig. R3-2a and 2d) could be fitted by six components (detailed peak information is shown in Table R3-2), corresponding to Nb-C, Nb⁺²-O, Nb⁺³-O, Nb⁺⁴-O, Nb₂O₅ and Nb-F, respectively. Fig. R3-2b and 2e display the C 1s regions of Nb₄C₃T_x and LB-Nb₄C₃T_x that are deconvoluted into five peaks, corresponding to Nb-C, C-C, CH_x, C-O, and -COO, respectively. Fig. R3-2c and 2f display the O 1s regions which are deconvoluted into five peaks, corresponding to Nb₂O₅, Nb-O_x, Nb-OH, C-O and H₂O, respectively. Similar to the phenomenon of LB-Ti₃C₂T_x, the strength of C-C and C-O bonds increases. Moreover, the F termination is further confirmed by C-Nb-F_x bond in F 1s region (Fig. R3-2g). After Lewis basic halides treatment, the Nb-F_x bond decreased due to the substitution of F termination by Br termination.

Mo₂Ti₂C₃T_x and LB-Mo₂Ti₂C₃T_x

In the high resolution XPS spectra of Mo₂Ti₂C₃T_x and LB-Mo₂Ti₂C₃T_x, the Mo 3d region (Fig. R3-3a and 3d) could be fitted by three components (detailed peak information is shown in Table R3-3). These components are mainly attributed to C-Mo and two small species (Mo⁺⁵ and Mo⁺⁶) belonging to mixed molybdenum oxides. The Ti 2p region (Fig. R3-3b and 3e) could be fitted by four components, corresponding to Ti-C, Ti⁺²-C, Ti⁺³-C and Ti₂O, respectively. Fig. R3-3c and 3f display the O 1s regions that are deconvoluted into five peaks, corresponding to MoO_x, Mo-O_x, Mo-OH, C-O and H₂O, respectively. The C 1s regions (Fig. R3-3g and 3h) of Mo₂Ti₂C₃T_x and LB-Mo₂Ti₂C₃T_x show five peaks, which could be fitted into Mo-C, C-C, CH_x, C-O, and -COO, respectively. Similar to the phenomenon of LB-Ti₃C₂T_x and LB-Nb₄C₃T_x, the strength of C-C and C-O bonds increases. And the F termination is further confirmed by C-Mo-F_x bond in F 1s region (Fig. R3-3i). After Lewis basic halides treatment, the Mo-F_x bond decreased due to the substitution of F termination by Br termination.

Action: According to the reviewer's suggestion, we show the XPS spectra of all elements and the corresponding analyses in the revised Supplementary Information:

Fig. R3-1. High resolution XPS spectra of **(a, d)** Ti 2p, **(b, e)** C 1s and **(c, f)** O 1s of $\text{Ti}_3\text{C}_2\text{T}_x$ and $\text{LB-Ti}_3\text{C}_2\text{T}_x$. (Fig. S6 in the revised manuscript)

In the high resolution XPS spectra of $\text{Ti}_3\text{C}_2\text{T}_x$ and $\text{LB-Ti}_3\text{C}_2\text{T}_x$, the Ti 2p region (Fig. S6a and 6d) could be fitted by five components (detailed peak information is shown in Table S2). The largest fractions are attributed to Ti-C bond, and smaller fractions of TiO_2 is also found. After treated by Lewis basic halides, $\text{LB-Ti}_3\text{C}_2\text{T}_x$ exhibits slight increase of TiO_2 and decrease of Ti-F bond, which could be induced by the substitution of F termination by Br termination. As shown in Fig. S6b and 6e, the C 1s regions of $\text{Ti}_3\text{C}_2\text{T}_x$ and $\text{LB-Ti}_3\text{C}_2\text{T}_x$ show five peaks, which could be fitted into Ti-C, C-C, CH_x , C-O, and -COO, respectively. The O 1s region (Fig. S6c and 6f) could be assigned to five peaks corresponding to TiO_2 , Ti-O_x, Ti-OH, C-O and H_2O , respectively. After Lewis basic halides treatment, the peaks of Ti-O_x and Ti-OH retained, indicating that O termination wasn't substituted by -Br. It's worth noting that the strength of C-C and C-O bonds increases in C 1s and O 1s regions of $\text{LB-Ti}_3\text{C}_2\text{T}_x$. Considering anhydrous diethyl ether and tetrahydrofuran containing C-C and C-O bonds are used for removing salts in our experiment, it is reasonable to speculate that there is a small amount of organic solvent residue on the surface of $\text{LB-Ti}_3\text{C}_2\text{T}_x$ in the process of washing salt.

Table R3-1. XPS fitting results of $\text{Ti}_3\text{C}_2\text{T}_x$ and LB- $\text{Ti}_3\text{C}_2\text{T}_x$. (Table S2 in revised manuscript)

$\text{Ti}_3\text{C}_2\text{T}_x$	BE (eV)	FWHM (eV)	Fraction	Assigned to
Ti $2p_{3/2}$ (Ti $2p_{1/2}$)	455.0 (461.0)	1.20 (1.20)	29.48	(O/OH)-Ti-C
	455.8 (461.6)	1.50 (1.50)	38.42	(O/OH)-Ti ⁺² -C
	456.9 (462.5)	1.32 (1.32)	17.61	(O/OH)-Ti ⁺³ -C
	458.8 (463.6)	1.70 (1.70)	9.34	TiO ₂
	460.3 (464.2)	1.10 (1.10)	5.15	Ti-F
C 1s	281.8	0.71	28.50	Ti-C
	284.8	1.43	48.56	C-C
	285.3	1.05	16.76	CH _x
	286.2	1.16	5.59	C-O
	289.1	1.00	0.59	-COO
O 1s	529.7	1.01	27.88	TiO ₂
	530.7	1.20	18.43	Ti-O _x
	531.7	1.25	12.76	Ti-OH
	532.6	1.09	6.32	C-O
	534.0	2.74	34.61	H ₂ O

LB-Ti₃C₂T_x	BE (eV)	FWHM (eV)	Fraction	Assigned to
Ti 2p _{3/2} (Ti 2p _{1/2})	454.9 (461.1)	1.00 (1.00)	20.73	(O/OH and Br)-Ti-C
	455.7 (461.7)	1.40 (1.40)	40.58	(O/OH and Br)-Ti ⁺² -C
	456.9 (462.7)	1.60 (1.60)	24.07	(O/OH and Br)-Ti ⁺³ -C
	459.2 (464.3)	1.60 (1.60)	13.24	TiO ₂
	460.4 (465.2)	1.00 (1.00)	1.38	Ti-F
C 1s	281.9	1.00	10.10	Ti-C
	284.8	1.20	51.44	C-C
	285.2	1.00	22.41	CH _x
	285.9	1.40	14.30	C-O
	289.2	0.92	1.75	-COO
O 1s	529.8	0.80	9.65	TiO ₂
	530.8	1.17	20.71	Ti-O _x
	531.8	1.17	17.09	Ti-OH
	532.7	1.34	39.27	C-O
	533.6	1.11	13.28	H ₂ O
Br 3d	68.9 (70.2)	1.56 (1.56)	1.00	Ti-Br

BE: Binding Energy

Fig. R3-2. High resolution XPS spectra of (a, d) Nb 3d, (b, e) C 1s, (c, f) O 1s and (g) F 1s of $\text{Nb}_4\text{C}_3\text{T}_x$ and $\text{LB-Nb}_4\text{C}_3\text{T}_x$. (Fig. S12 in revised manuscript)

In the high resolution XPS spectra of $\text{Nb}_4\text{C}_3\text{T}_x$ and $\text{LB-Nb}_4\text{C}_3\text{T}_x$, the Nb 3d region (Fig. S12a and 12d) could be fitted by six components (detailed peak information is shown in Table S4), corresponding to Nb-C, $\text{Nb}^{+2}\text{-O}$, $\text{Nb}^{+3}\text{-O}$, $\text{Nb}^{+4}\text{-O}$, Nb_2O_5 and Nb-F, respectively. Fig. S12b and 12e display the C 1s regions of $\text{Nb}_4\text{C}_3\text{T}_x$ and $\text{LB-Nb}_4\text{C}_3\text{T}_x$ that are deconvoluted into five peaks, corresponding to Nb-C, C-C, CH_x , C-O, and -COO, respectively. Fig. S12c and 12f display the O 1s regions which are deconvoluted into five peaks, corresponding to Nb_2O_5 , Nb-O_x , Nb-OH, C-O and H_2O , respectively. Similar to the phenomenon of $\text{LB-Ti}_3\text{C}_2\text{T}_x$, the strength of C-C and C-O bonds increases. Moreover, the F termination is further confirmed by C-Nb- F_x bond in F 1s region (Fig. S12g). After Lewis basic halides treatment, the Nb- F_x bond decreased due to the substitution of F termination by Br termination.

Table R3-2. XPS fitting results of Nb₄C₃T_x and LB-Nb₄C₃T_x. (Table S4 in revised manuscript)

Nb ₄ C ₃ T _x	BE (eV)	FWHM (eV)	Fraction	Assigned to
Nb 3d _{5/2} (Nb 3d _{3/2})	203.6 (206.4)	0.87 (0.87)	33.40	Nb-C
	204.4 (207.1)	1.07 (1.07)	32.96	Nb ⁺² -O
	205.3 (207.9)	0.97 (0.97)	9.79	Nb ⁺³ -O
	206.1 (209.0)	1.10 (1.10)	10.76	Nb ⁺⁴ -O
	207.4 (210.0)	1.31 (1.31)	8.84	Nb ₂ O ₅
	208.3 (210.9)	1.41 (1.41)	4.25	Nb-F
C 1s	282.7	0.9	26.86	C-Nb
	284.8	1.22	43.7	C-C
	285.2	1.62	24.7	CH _x
	286.5	1.5	2.15	C-O
	289.0	1.39	2.59	-COO
O 1s	530.1	1.00	40.57	Nb ₂ O ₅
	530.9	1.18	22.45	Nb-O _x
	531.9	1.28	21.65	Nb-OH
	533.0	1.03	7.27	C-O
	533.8	1.98	8.06	H ₂ O
F 1s	684.2	1.04	52.04	C-Nb-F _x
	685.0	1.2	30.89	AlF _x
	686.6	1.56	17.07	C-F

LB-Nb₄C₃T_x	BE (eV)	FWHM (eV)	Fraction	Assigned to
Nb 3d _{5/2} (Nb 3d _{3/2})	203.6 (206.4)	0.99 (0.99)	41.59	Nb-C
	204.4 (207.0)	1.00 (1.00)	24.54	Nb ⁺² -O
	205.2 (208.1)	1.10 (1.10)	11.32	Nb ⁺³ -O
	206.1 (208.8)	1.07 (1.07)	8.51	Nb ⁺⁴ -O
	207.4 (209.7)	1.05 (1.05)	10.51	Nb ₂ O ₅
	207.8 (210.6)	1.27 (1.27)	3.53	Nb-F
C 1s	282.8	1.03	28.28	C-Nb
	284.8	1.40	41.93	C-C
	285.3	1.40	22.13	CH _x
	286.5	1.32	7.16	C-O
	289.0	1.26	0.50	-COO
O 1s	530.2	1.20	38.44	Nb ₂ O ₅
	531.0	0.86	14.14	Nb-O _x
	531.8	0.91	13.42	Nb-OH
	532.6	1.35	28.96	C-O
	533.6	0.90	5.04	H ₂ O
F 1s	684.4	1.22	37.74	C-Nb-F _x
	685.2	1.75	50.12	AlF _x
	686.7	1.01	12.14	C-F
Br 3d _{5/2} (Br 3d _{3/2})	68.6 (69.5)	1.37 (1.37)	1.00	Nb-Br

Fig. R3-3. High resolution XPS spectra of (a, d) Mo 3d, (b, e) Ti 2p, (c, f) O 1s, (g, h) C 1s and (i) F 1s of $\text{Mo}_2\text{Ti}_2\text{C}_3\text{T}_x$ and $\text{LB-Mo}_2\text{Ti}_2\text{C}_3\text{T}_x$. (Fig. S14 in revised manuscript)

In the high resolution XPS spectra of $\text{Mo}_2\text{Ti}_2\text{C}_3\text{T}_x$ and $\text{LB-Mo}_2\text{Ti}_2\text{C}_3\text{T}_x$, the Mo 3d region (Fig. S14a and 14d) could be fitted by three components (detailed peak information is shown in Table S5). These components are mainly attributed to C-Mo and two small species (Mo^{+5} and Mo^{+6}) belonging to mixed molybdenum oxides. The Ti 2p region (Fig. S14b and 14e) could be fitted by four components, corresponding to Ti-C, Ti^{+2} -C, Ti^{+3} -C and Ti_2O , respectively. Fig. S14c and 14f display the O 1s regions that are deconvoluted into five peaks, corresponding to MoO_x , Mo-O_x , Mo-OH, C-O and H_2O , respectively. The C 1s regions (Fig. S14g and 14h) of $\text{Mo}_2\text{Ti}_2\text{C}_3\text{T}_x$ and $\text{LB-Mo}_2\text{Ti}_2\text{C}_3\text{T}_x$ show five peaks, which could be fitted into Mo-C, C-C, CH_x , C-O, and -COO, respectively. Similar to the phenomenon of $\text{LB-Ti}_3\text{C}_2\text{T}_x$ and $\text{LB-Nb}_4\text{C}_3\text{T}_x$, the strength of C-C and C-O bonds increases. And the F termination is further confirmed by C-Mo-F_x bond in F 1s region (Fig. S14i). After Lewis basic halides treatment, the

Mo-F_x bond decreased due to the substitution of F termination by Br termination.

Table R3-3. XPS fitting results of Mo₂Ti₂C₃T_x and LB-Mo₂Ti₂C₃T_x. (Table S5 in revised manuscript)

Mo ₂ Ti ₂ C ₃ T _x	BE (eV)	FWHM (eV)	Fraction	Assigned to
Mo 3d _{5/2} (Mo 3d _{3/2})	229.1 (232.2)	0.95 (0.95)	66.99	C-Mo-T _x
	231.4 (234.8)	1.89 (1.89)	19.58	Mo ⁺⁵
	232.8 (235.6)	1.45 (1.45)	13.43	Mo ⁺⁶
Ti 2p _{3/2} (Ti 2p _{1/2})	455.1 (460.8)	1.13 (1.13)	60.89	Ti-C
	456.1 (461.5)	1.36 (1.36)	20.22	Ti ⁺² -C
	457.6 (462.2)	1.71 (1.71)	9.68	Ti ⁺³ -C
	459.3 (463.5)	1.65 (1.65)	9.21	TiO ₂
C 1s	282.3	1.22	31.31	C-Mo
	284.8	1.27	33.73	C-C
	285.2	1.33	19.35	CH _x
	286.3	1.31	15.12	C-O
	288.4	1.00	0.49	-COO
O 1s	530.0	1.13	44.68	MnO _x
	530.9	1.08	19.27	Mn-O _x
	532.0	1.63	22.23	Mn-OH
	532.7	1.16	7.50	C-O
	533.7	1.53	6.32	H ₂ O
F 1s	684.6	2.2	16.61	C-Mo-F _x
	687.0	1.84	19.10	AlF _x
	689.8	1.79	65.29	C-F

LB-Mo ₂ Ti ₂ C ₃ T _x	BE (eV)	FWHM (eV)	Fraction	Assigned to
Mo 3d _{5/2} (Mo 3d _{3/2})	229.3 (232.4)	1.01 (1.01)	78.80	C-Mo-T _x
	231.6 (235.0)	1.40 (1.40)	10.36	Mo ⁺⁵
	233.1 (235.8)	1.45 (1.45)	10.84	Mo ⁺⁶
Ti 2p _{3/2} (Ti 2p _{1/2})	455.3 (461.1)	1.13 (1.13)	51.80	Ti-C
	456.1 (461.8)	1.66 (1.66)	28.39	Ti ⁺² -C
	458.0 (462.5)	1.75 (1.75)	9.70	Ti ⁺³ -C
	459.5 (463.7)	1.75 (1.75)	10.11	TiO ₂
C 1s	282.6	1.22	23.03	C-Mo-T _x
	284.8	1.18	46.87	C-C
	285.3	1.00	13.12	CH _x
	286.4	1.46	16.33	C-O
	288.5	0.50	0.65	-COO
O 1s	530.3	1.26	36.64	MnO _x
	531.2	0.90	10.18	Mn-O _x
	531.8	0.82	9.02	Mn-OH
	532.6	1.56	37.64	C-O
	533.7	1.12	6.52	H ₂ O
F 1s	685.0	2.12	16.82	C-Mo-F _x
	687.3	2.50	19.45	AlF _x
	689.9	1.59	63.73	C-F
Br 3d _{5/2} (Br 3d _{3/2})	69.3 (70.1)	1.68 (1.68)	1.00	Mo-Br

Comment 2. *The authors are strongly emphasizing the expanded interlayer space but attribute the increase in d to the wrong reasons. Higher, d-spacings have been achieved for water etched MXenes as well. So there is nothing special here.*

Response: We would like to thank for the reviewer's comments and totally understand

the concern. We didn't intend to emphasize the expanded interlayer space, instead, we would like to show that our method could tune the termination and interlayer spacing simultaneously, especially for MXenes besides $\text{Ti}_3\text{C}_2\text{T}_x$. Following reviewer's suggestion, we revised the statement in the manuscript and further explained the reasons of enlarged interlayer spacing, which is attributed to the synergism of termination substitution and desolvated cations intercalation.

Action: We have modified the sentences in our revised manuscript, page 1 and 7:

“This is in accordance with the variation trend of the amount of desolvated Na^+ and K^+ in $\text{AlBr}_3/\text{NaBr}/\text{KBr}$ molten salts as mentioned previously, demonstrating the increased interlayer spacing of $\text{Ti}_3\text{C}_2\text{T}_x$ is dominantly attributed to the efficient intercalation of desolvated Na^+ and K^+ from $\text{AlBr}_3/\text{NaBr}/\text{KBr}$.”

Comment 3. The authors claim: "If the above two processes occur in aqueous solution, the overall activation energy increases due to the energy consumption for the necessary step of ion de-solvation, which limits the reaction rate according to the Arrhenius equation" This is simply not true. In aqueous solutions, solvated ions readily intercalate at room temperature. That's how one obtains 16 Å spacing.

Response: We agree with reviewer that solvated ions readily intercalate at RT. Following reviewer's suggestion, now we would like to emphasize that the desolvated ions require a lower activation energy than the solvated ions in the termination substitution process according to the Arrhenius equation. This is evidential that the substitution reaction of Br termination cannot be carried out in aqueous solution. At the same time, the desolvated cations could intercalate into the narrow interlayer space of MXenes (besides $\text{Ti}_3\text{C}_2\text{T}_x$) to efficiently increase the distance. This is the advantage of desolvated ions in our method.

Action: According to reviewer's comment, we have modified and marked the sentences in our revised manuscript, page 3:

“If the above termination substitution process occurs in aqueous solution, the overall activation energy increases due to the energy consumption for the necessary step of ion de-solvation, which limits the reaction rate according to the Arrhenius equation. At the same time, the desolvated cations could intercalate into the narrow

interlayer space of MXenes to efficiently increase the distance.”

Comment 4. *The authors claim, " In this context, the molten salt system is of particular interest because “naked” ions could be directly involved in reaction, which is beneficial for largely increasing interlayer spacing and efficient termination substitution." I don't get logic here. If higher interlayer spacings are beneficial, then ideally one should prefer solvated ions over naked ions as the former have larger diameters than the latter.*

Response: We think our expression is not appropriate here. The diameter of the solvated ions is indeed larger than that of the naked ions, and 16 Å spacing was achieved in $Ti_3C_2T_x$ MXene at wet state. However, it is still difficult to enlarge the interlayer spacing of $Nb_4C_3T_x$ and $Mo_2Ti_2C_3T_x$ via intercalation in salt solutions (NaCl, LiCl). By contrast, in our work, the “naked” ions are easier to intercalate into $Nb_4C_3T_x$ and $Mo_2Ti_2C_3T_x$. Therefore, we changed the statement to express that the “naked” anions are beneficial for termination substitution and “naked” cations could simultaneously intercalate MXenes.

Action: According to reviewer’s comment, we have modified the sentences in our revised manuscript, page 3 and 7:

“In this context, the molten salt system is of particular interest as “naked” anions are beneficial for termination substitution and “naked” cations could simultaneously intercalate MXenes”

“For short conclusion, the desolvated anions (Br^-) in $AlBr_3/NaBr/KBr$ play the important role on substituting termination and desolvated cations (Na^+ and K^+) could simultaneously enlarge the interlayer spacing.”

Comment 5. *The authors claim, "Such a large interlayer spacing is rarely achievable via the intercalation of inorganic cations, as interlayer spacing ranging between 12.3 Å and 13.6 Å is generally reported for Li^+ intercalated multilayer $Ti_3C_2T_x$ " This increase in d is mostly because of -Br or -I terminations and not the cations perse." Here again it appears the authors are really not familiar with the literature on interlayer spacings in MXenes. In the LiF/HCl method 16 Å is routine.*

Response: Thanks for reviewer’s comments. Here we notice that the interlayer spacing

of 16 Å is achieved in the wet samples (*Chem. Mater.* 2022, 34, 678-693), while our samples are adequately dried in vacuum oven. We have performed LiF/HCl etching Ti_3AlC_2 experiment, and the obtained $\text{Ti}_3\text{C}_2\text{T}_x$ was dried in vacuum at 60 °C for 12 h. The interlayer spacing of multilayer $\text{Ti}_3\text{C}_2\text{T}_x$ (LiF/HCl) calculated by XRD is 12.9 Å (6.9°). Furthermore, we have summarized the literature on ion intercalation and listed the information of interlayer spacing in Table R3-4.

We highly appreciate the comments from reviewer and reconsider the reasons of enlarged interlayer spacing. Kamysbayev, V. et al. reported that the interlayer spacing changes with the varied termination species (*Science* 2020, 369, 979-983). But we can find that the changes are slight. Moreover, the change trend of interlayer spacing is linear which is consistent with the change of the content of desolvated cations in Lewis basic halides (the change trend of anions is exponential), indicating cations contribute dominantly in expanding the interlayer spacing. To make it clear, we change the statement to emphasize the synergistic effect of Br substitution and cation intercalation on enlarging interlayer spacing.

Action: We have modified the sentences in our revised manuscript, page 1, 3, 4, 7 and 11:

“Benefited from the abundant desolvated halogen anions and cations in molten state Lewis-basic halides, the -F termination was substituted by nucleophilic reaction and the interlayer spacing was enlarged.”

“Such a large interlayer spacing is superior to that of many reported value of MXenes intercalated by various cation species. The interlayer spacing data of MXenes are summarized in Table S1.”

“As shown in Fig. 1, the interlayer spacing of MXene could be enlarged due to the synergism of termination substitution and desolvated cations intercalation (Na^+ and K^+) in Lewis-basic halides (Fig. 1a)”

“The desolvated halogen ions are involved in the termination substitution via nucleophilic reaction. On the other hand, the synergistic effect of termination substitution and cations (abundant desolvated Na^+ and K^+) intercalation are responsible for the enlarged interlayer spacing.”

“This is in accordance with the variation trend of the amount of desolvated Na⁺ and K⁺ in AlBr₃/NaBr/KBr molten salts as mentioned previously, demonstrating the increased interlayer spacing of Ti₃C₂T_x is dominantly attributed to the efficient intercalation of desolvated Na⁺ and K⁺ from AlBr₃/NaBr/KBr.”

Table R3-4. The interlayer spacing of multilayer MXene after intercalation. (Table S1 in revised manuscript)

Intercalating solution	(Multilayer MXene) Interlayer spacing	Ref.
LiF/HCl	13.50-14.00 Å	1
LiF/HCl	12.42 Å	2
MgCl ₂	14.29 Å	3
NaCl	11.85 Å	3
N-butyllithium	12.38 Å	4
LiOH	12.38 Å	4
LiOH	13.60 Å	5
SnCl ₄	12.42 Å	6
LiCl	11.50 Å	6
NaCl	11.01 Å	6
N ₂ H ₄ ·H ₂ O	12.70 Å	7
KOH	12.50 Å	8
LiF/HCl	12.90 Å	This work
NaBr, KBr	14.60 Å	This work

Reference

1. Ghidui, M., Lukatskaya, M. R., Zhao, M. Q., Gogotsi, Y. & Barsoum, M. W. Conductive two-dimensional titanium carbide 'clay' with high volumetric capacitance. *Nature* 516, 78-81 (2014).
2. Lipatov, A. et al. Effect of synthesis on quality, electronic properties and environmental stability

- of individual monolayer Ti_3C_2 MXene flakes. *Adv. Electron. Mater.* 1600255 (2016).
3. Al-Temimy, A. et al. Impact of cation intercalation on the electronic structure of $\text{Ti}_3\text{C}_2\text{T}_x$ MXenes in sulfuric acid. *ACS Appl. Mater. Interfaces* 12, 15087-15094 (2020).
 4. Chen, X. F. et al. N-butyllithium-treated $\text{Ti}_3\text{C}_2\text{T}_x$ MXene with excellent pseudocapacitor performance. *ACS Nano* 13, 9449-9456 (2019).
 5. Wang, H. B. et al. Achieving high-rate capacitance of multi-layer titanium carbide (MXene) by liquid-phase exfoliation through Li-intercalation. *Electrochem. Commun.* 81, 48-51 (2017).
 6. Hu, A. L., Y, J., Zhao, H. Z., Zhang, H., Li, W. One-step synthesis for cations intercalation of two-dimensional carbide crystal Ti_3C_2 MXene. *Appl. Surf. Sci.* 505, 144538 (2020).
 7. Mashtalir, O. et al. Effect of hydrazine intercalation on structure and capacitance of 2D titanium carbide (MXene). *Nanoscale* 8, 9128-9133 (2016).
 8. Li, J. et al. Achieving high pseudocapacitance of 2D titanium carbide (MXene) by cation intercalation and surface modification. *Adv. Energy Mater.* 7, 1602725 (2017).

Comment 6. *Along the same lines the hype here is unwarranted. The authors claim "Although numerous efforts have been devoted to the regulation of $\text{Ti}_3\text{C}_2\text{T}_x$ MXene, finding a general and facile route to tuning interlayer spacing and surface termination of various MXenes remains challenging while demanded." And then proceed to describe a process where the MAX phase needs first to be etched with HF, then placed in one molten bath, taken out of it, washing the powders and then placed in a second molten bath, before taking it out and washing it, etc. Whatever you call this, it is not facile by any stretch of the imagination. Their dismissal of previous work is also off base. If I compare their method to say DMSO all carried out in water at RT, there is no doubt which is the more facile. Similarly, dismissing previous molten salt as being too high a temperature is also off base.*

Response: We understand the concern from reviewer and really appreciate the suggestions. Actually, we didn't intend to overlook the previous work. We would like to show the advantage of our method: first, our method required a low temperature which can be obtained with a normal magnetic stirring heater easily. As a comparison,

high temperature (450~750°C) is needed in previous works (Table R3-5) which is conducted by tube furnace (*Nat. Mater.* 2021, 19, 894-899) or muffle furnace (*Science* 2020, 369, 979-983). Second, our method is capable of simultaneously tuning the termination and interlayer spacing, especially for MXenes besides $Ti_3C_2T_x$. Following reviewer's suggestion, we tone down the statement (for example, we deleted the statement of "finding a general and facile route" et al.) and show more respect to the previous work.

Table R3-5. The interlayer spacing and temperature of molten salt etching or modifying method of MXene.

Molten Salt	Temperature	Interlayer spacing (Multilayer MXene)	Reference
NaCl KCl CuCl ₂ (Ti ₃ C ₂ T _x)	700°C (10-40) mins	(9.3 to 11.1 Å)	1
NaCl KCl CuCl ₂ (Ti ₃ C ₂ T _x)	680°C	(9.17 to 10.98 Å)	2
NaCl KCl CuCl ₂ (Nb ₂ C)	750°C	(13.8 to 17.7 Å)	3
CuCl ₂	750°C	(8.8 to 10.98 Å)	4
CuCl ₂ /CoCl ₂	750°C	(9.1 to 10.85 Å)	5
LiCl-KCl K ₂ CO ₃	450°C	(9.6 to 10.5~12.1 Å)	6
AlBr ₃ /NaBr/KBr	230°C	(11.2 to 14.6 Å)	This work

Reference

- Chen, J. J. et al. Molten salt-shielded synthesis (MS³) of MXenes in Air. *Energy Environ. Mater.* 0, 1-6 (2022).
- Liu, L. Y. et al. Exfoliation and delamination of Ti₃C₂T_x MXene prepared via molten salt etching route. *ACS Nano* 16, 111-118 (2022).
- Dong, H. Y. et al. Molten salt derived Nb₂CT_x MXene anode for Li-ion batteries. *Chem. Electro. Chem.* 8, 957-962 (2021).
- Li, Y. B. et al. A general Lewis acidic etching route for preparing MXenes with enhanced electrochemical performance in non-aqueous electrolyte. *Nat. Mater.* **19**, 894-899 (2021).
- Bai, Y. et al. MXene-copper/cobalt hybrids via lewis acidic molten salts etching for high performance symmetric supercapacitors. *Angew. Chem.* 133, 25522 -25526 (2021).
- Luo, G. et al. High capacitance of MXene (Ti₃C₂T_x) through intercalation and surface modification in molten salt. *Electrochim. Acta* doi.org/10.1016/j.electacta.2021.139476. (2021).

Action: We have modified the sentences in our revised manuscript, page 1 and 2:

“Numerous efforts have been devoted to the regulation of $\text{Ti}_3\text{C}_2\text{T}_x$ MXene, however, tuning interlayer spacing and surface halogen termination of other MXenes (besides $\text{Ti}_3\text{C}_2\text{T}_x$) remains challenging while demanded.”

“For Lewis acid molten salt method (above 450 °C), termination is more controllable and tunable”

“However, in molten salt system, the resulted MXenes showed a relatively small interlayer spacing (about 10.9 Å), in which the interlayer spacing could only be enlarged by organic reagents (difficult to be removed) intercalation in previous researches.”

Comment 7. If I look at Table S1, there is serious reduction in the Ti-C MXene signal. That is worrisome. DO the authors have an explanation?

Response: We noted that, in Ti 2p region of XPS, the proportion of Ti-C decreased from 85.51% to 85.38% (almost no decrease), which showed that the $\text{Ti}_3\text{C}_2\text{T}_x$ was not destroyed. However, the signal of Ti-C MXene was relatively decreased in C 1s. Therefore, we infer that there is a small amount of organic solvent residue (anhydrous diethyl ether and tetrahydrofuran) on the surface of MXene in the process of washing salt, which leads to the increase of C-C and C-O signals. The increase of C-O signal can also be proved by O 1s region of LB- $\text{Ti}_3\text{C}_2\text{T}_x$ MXene.

Comment 8. Why are the authors not showing the Nb results in Table S3?

Response: Based on the reviewer’s suggestion, we have showed Nb results in Table R3-2.

Action: We have added the Table S4 in Supplementary Information.

Table R3-2. XPS fitting results of $\text{Nb}_4\text{C}_3\text{T}_x$ and LB- $\text{Nb}_4\text{C}_3\text{T}_x$. (Table S4 in revised manuscript)

Nb₄C₃T_x	BE (eV)	FWHM (eV)	Fraction	Assigned to
Nb 3d _{5/2} (Nb 3d _{3/2})	203.6 (206.4)	0.87 (0.87)	33.40	Nb-C
	204.4 (207.1)	1.07 (1.07)	32.96	Nb ⁺² -O
	205.3 (207.9)	0.97 (0.97)	9.79	Nb ⁺³ -O
	206.1 (209.0)	1.10 (1.10)	10.76	Nb ⁺⁴ -O
	207.4 (210.0)	1.31 (1.31)	8.84	Nb ₂ O ₅
	208.3 (210.9)	1.41 (1.41)	4.25	Nb-F
C 1s	282.7	0.9	26.86	C-Nb
	284.8	1.22	43.7	C-C
	285.2	1.62	24.7	CH _x
	286.5	1.5	2.15	C-O
	289.0	1.39	2.59	-COO
O 1s	530.1	1.00	40.57	Nb ₂ O ₅
	530.9	1.18	22.45	Nb-O _x
	531.9	1.28	21.65	Nb-OH
	533.0	1.03	7.27	C-O
	533.8	1.98	8.06	H ₂ O
F 1s	684.2	1.04	52.04	C-Nb-F _x
	685.0	1.2	30.89	AlF _x
	686.6	1.56	17.07	C-F

LB-Nb ₄ C ₃ T _x	BE (eV)	FWHM (eV)	Fraction	Assigned to
Nb 3d _{5/2} (Nb 3d _{3/2})	203.6 (206.4)	0.99 (0.99)	41.59	Nb-C
	204.4 (207.0)	1.00 (1.00)	24.54	Nb ⁺² -O
	205.2 (208.1)	1.10 (1.10)	11.32	Nb ⁺³ -O
	206.1 (208.8)	1.07 (1.07)	8.51	Nb ⁺⁴ -O
	207.4 (209.7)	1.05 (1.05)	10.51	Nb ₂ O ₅
	207.8 (210.6)	1.27 (1.27)	3.53	Nb-F
C 1s	282.8	1.03	28.28	C-Nb
	284.8	1.40	41.93	C-C
	285.3	1.40	22.13	CH _x
	286.5	1.32	7.16	C-O
	289.0	1.26	0.50	-COO
O 1s	530.2	1.20	38.44	Nb ₂ O ₅
	531.0	0.86	14.14	Nb-O _x
	531.8	0.91	13.42	Nb-OH
	532.6	1.35	28.96	C-O
	533.6	0.90	5.04	H ₂ O
F 1s	684.4	1.22	37.74	C-Nb-F _x
	685.2	1.75	50.12	AlF _x
	686.7	1.01	12.14	C-F
Br 3d _{5/2} (Br 3d _{3/2})	68.6 (69.5)	1.37 (1.37)	1.00	Nb-Br

Comment 9. *Why did the authors use tetrahydrofuran and diethyl ether for washing?*

Response: Dissolving AlBr₃ in water will generate a lot of heat, and the reaction is violent. The solubility of AlBr₃ by anhydrous diethyl ether is larger, and the exothermic heat is smaller. Next, the solubility of anhydrous diethyl ether in water is small (slightly soluble), and tetrahydrofuran is used to wash anhydrous diethyl ether (*Nature* 2017, 542, 328-331).

REVIEWER COMMENTS

Reviewer #1 (Remarks to the Author):

I appreciate the authors' careful responses to the other reviewers' and my comments. I believe the manuscript has been strengthened appreciably and is now suitable for publication.

Reviewer #2 (Remarks to the Author):

The author mentioned what I care about. I think the current status is acceptable.

Reviewer #4 (Remarks to the Author):

The authors have added XPS as requested and also made some other changes. But I still have some comments on the revised manuscript.

1) In the XPS discussion of -Br substituted Ti₃C₂ the authors say " It's worth noting that the strength of C-C and C-O bonds increases in C 1s and O 1s regions of LB-Ti₃C₂T_x". This is incorrect. An increase in peak area corresponds to an increase in the % content of the species, it does not imply, an increase in bond strength as the authors say.

2) XPS fittings need appropriate citations. How did the authors attribute the B.E's to specific components? give reference for each.

3) Why is there FWHM change for the same components before and after -Br substitution?

4) The authors need to fit the MXene contributions in XPS with asymmetric peaks, using symmetric peaks causes unwanted extra components. For ex: in the case of Mo 3d peak in Fig S14 and d, the contribution from oxide will be much lower than shown. See some recent reviews on XPS of MXenes, they discuss this exact problem.

5) Please quantify the chemical formulas based on XPS before and after salt treatment.

6) The authors say " At the same time, the desolvated cations could intercalate into the narrow interlayer space of MXenes to efficiently increase the distance". Ions always intercalate between MXene layers to balance charge. So there is nothing special per say here that the authors are emphasizing. De-solvated ions can be achieved in regular MXene by mere vacuum drying, and even in this case, will the desolvated ions not solvate as soon as they come in contact with the electrolyte in supercapacitors?

7) " For short conclusion, the desolvated anions (Br⁻) in AlBr₃/NaBr/KBr play an important role on substituting termination and desolvated cations (Na⁺ and K⁺) could simultaneously enlarge the interlayer spacing." I think the authors are missing out on the point that both, the ion and the -Br contribute to increased interlayer spacing. Also, I still don't get what's so special about naked ion intercalation.

8) In response to Comment 5, the authors need to compare their d-spacings with MXene synthesised in a mix of HI+HCl or HBr+HCl for true comparison. There are few papers in the literature that do that.

9) The authors say " Numerous efforts have been devoted to the regulation of Ti₃C₂T_x MXene, however, tuning interlayer spacing and surface halogen termination of other MXenes (besides

Ti3C2Tx) remains challenging while demanded". I disagree with the fact that its challenging. It is for most part lack of motivation to try what works for Ti3C2 as is on other MXenes when there is no apparent benefit of using them over Ti3C2.

Response Notes

(Simultaneously tuning interlayer spacing and termination of MXenes by Lewis-basic halides)

We appreciate for the editor and reviewers' constructive suggestions to our manuscript. All the suggestions are very helpful for us to improve our research. We have addressed all comments of the reviewers, and our responses to the reviewer's comments are listed below point-by-point. The revised sections have been highlighted in the revised manuscript.

Reviewer #1

I appreciate the authors' careful responses to the other reviewers' and my comments. I believe the manuscript has been strengthened appreciably and is now suitable for publication.

Response: We appreciate the efforts the reviewer has spent on our manuscript. Thanks a lot for all the positive comments on our work.

Reviewer #2

The author mentioned what I care about. I think the current status is acceptable.

Response: We appreciate the efforts the reviewer has spent on our manuscript and thanks for the reviewer's positive comments.

Reviewer #4

The authors have added XPS as requested and also made some other changes. But I still have some comments on the revised manuscript.

Response: We appreciate the efforts the reviewer has spent on our manuscript and thanks for the reviewer's positive comments. We have further revised the manuscript per reviewer's suggestion. The point-by-point response has been done as follows.

Comment 1. In the XPS discussion of -Br substituted Ti_3C_2 the authors say " It's worth noting that the strength of C-C and C-O bonds increases in C 1s and O 1s regions of LB- $Ti_3C_2T_x$ ". This is incorrect. An increase in peak area corresponds to an increase in the % content of the species, it does not imply, an increase in bond strength as the authors say.

Response: We thank the reviewer for pointing this out. We have revised the related description.

Action: We have modified the sentence in our Supplementary Information, page 3:

“It's worth noting that the content of C-C and C-O bonds increases in C 1s and O 1s regions of LB-Ti₃C₂T_x”.

Comment 2. XPS fittings need appropriate citations. How did the authors attribute the B.E' s to specific components? give reference for each.

Response: Thanks for reviewer's comments and we have added the reference for XPS fittings. We attribute the B.E' s to specific components based on previous research and related literatures.

Action: We show the reference of XPS fittings in the revised Supplementary Information.

Table R4-1. XPS fitting results of Ti₃C₂T_x and LB-Ti₃C₂T_x. (Table S2 in revised manuscript)

Ti ₃ C ₂ T _x	BE (eV)	FWHM (eV)	Fraction	Assigned to	Ref.
Ti 2p _{3/2} (Ti 2p _{1/2})	455.0 (461.0)	1.20 (1.10)	29.48	(O/OH)-Ti-C	1
	455.8 (461.6)	1.50 (1.30)	38.42	(O/OH)-Ti ⁺² -C	1
	456.9 (462.5)	1.32 (1.60)	17.61	(O/OH)-Ti ⁺³ -C	1
	458.8 (463.6)	1.60 (1.70)	9.34	TiO ₂	1
	460.3 (464.2)	1.10 (1.20)	5.15	Ti-F	1
C 1s	281.8	0.71	28.50	Ti-C	2
	284.8	1.43	48.56	C-C	2
	285.3	1.05	16.76	CH _x	2
	286.2	1.16	5.59	C-O	2
	289.1	1.00	0.59	-COO	2
O 1s	529.7	1.01	27.88	TiO ₂	2
	530.7	1.20	18.43	Ti-O _x	2
	531.7	1.25	12.76	Ti-OH	2

	532.6	1.09	6.32	C-O	2
	534.0	2.74	34.61	H ₂ O	2

LB-Ti ₃ C ₂ T _x	BE (eV)	FWHM (eV)	Fraction	Assigned to	Ref.
Ti 2p _{3/2} (Ti 2p _{1/2})	454.9 (461.1)	1.00 (1.10)	20.73	(O/OH and Br)-Ti-C	1
	455.7 (461.7)	1.40 (1.30)	40.58	(O/OH and Br)-Ti ⁺² -C	1
	456.9 (462.7)	1.60 (1.65)	24.07	(O/OH and Br)-Ti ⁺³ -C	1
	459.2 (464.3)	1.60 (1.70)	13.24	TiO ₂	1
	460.4 (465.2)	1.00 (0.80)	1.38	Ti-F	1
C 1s	281.9	1.00	10.10	Ti-C	2
	284.8	1.20	51.44	C-C	2
	285.2	1.00	22.41	CH _x	2
	285.9	1.40	14.30	C-O	2
	289.2	0.92	1.75	-COO	2
O 1s	529.8	0.80	9.65	TiO ₂	2
	530.8	1.17	20.71	Ti-O _x	2
	531.8	1.17	17.09	Ti-OH	2
	532.7	1.34	39.27	C-O	2
	533.6	1.11	13.28	H ₂ O	2
Br 3d	68.9 (70.2)	1.56 (1.56)	1.00	Ti-Br	3

BE: Binding Energy

Table R4-2. XPS fitting results of Nb₄C₃T_x and LB-Nb₄C₃T_x. (Table S4 in revised manuscript)

Nb ₄ C ₃ T _x	BE (eV)	FWHM (eV)	Fraction	Assigned to	Ref.
Nb 3d _{5/2} (Nb 3d _{3/2})	203.6 (206.4)	0.89 (0.93)	33.40	Nb-C	4
	204.4 (207.1)	1.10 (1.14)	32.96	Nb ⁺² -O	4
	205.3 (207.9)	1.00 (1.04)	9.79	Nb ⁺³ -O	4

	206.1 (209.0)	1.14 (1.18)	10.76	Nb ⁺⁴ -O	4
	207.4 (210.0)	1.35 (1.38)	8.84	Nb ₂ O ₅	4
	208.3 (210.9)	1.18 (1.21)	4.25	Nb-F	4
C 1s	282.7	0.90	26.86	C-Nb	5
	284.8	1.22	43.7	C-C	5
	285.2	1.62	24.7	CH _x	5
	286.5	1.5	2.15	C-O	5
	289.0	1.39	2.59	-COO	5
O 1s	530.1	1.00	40.57	Nb ₂ O ₅	6
	530.9	1.18	22.45	Nb-O _x	6
	531.9	1.28	21.65	Nb-OH	6
	533.0	1.03	7.27	C-O	6
	533.8	1.98	8.06	H ₂ O	6
F 1s	684.2	1.04	52.04	C-Nb-F _x	6
	685.0	1.2	30.89	AlF _x	6
	686.6	1.56	17.07	C-F	6

LB-Nb ₄ C ₃ T _x	BE (eV)	FWHM (eV)	Fraction	Assigned to	Ref.
Nb 3d _{5/2} (Nb 3d _{3/2})	203.6 (206.4)	1.02 (1.05)	41.59	Nb-C	4
	204.4 (207.0)	1.03 (1.06)	24.54	Nb ⁺² -O	4
	205.2 (208.1)	1.14 (1.17)	11.32	Nb ⁺³ -O	4
	206.1 (208.8)	1.10 (1.13)	8.51	Nb ⁺⁴ -O	4
	207.4 (209.7)	1.08 (1.11)	10.51	Nb ₂ O ₅	4
	207.8 (210.6)	1.30 (1.34)	3.53	Nb-F	4
C 1s	282.8	1.03	28.28	C-Nb	5
	284.8	1.40	41.93	C-C	5
	285.3	1.40	22.13	CH _x	5
	286.5	1.32	7.16	C-O	5

	289.0	1.26	0.50	-COO	5
O 1s	530.2	1.20	38.44	Nb ₂ O ₅	6
	531.0	0.86	14.14	Nb-O _x	6
	531.8	0.91	13.42	Nb-OH	6
	532.6	1.35	28.96	C-O	6
	533.6	0.90	5.04	H ₂ O	6
F 1s	684.4	1.22	37.74	C-Nb-F _x	6
	685.2	1.75	50.12	AlF _x	6
	686.7	1.01	12.14	C-F	6
Br 3d _{5/2} (Br 3d _{3/2})	68.6 (69.5)	1.37 (1.37)	1.00	Nb-Br	7

Table R4-3. XPS fitting results of Mo₂Ti₂C₃T_x and LB-Mo₂Ti₂C₃T_x. (Table S5 in revised manuscript)

Mo ₂ Ti ₂ C ₃ T _x	BE (eV)	FWHM (eV)	Fraction	Assigned to	Ref.
Mo 3d _{5/2} (Mo 3d _{3/2})	229.1 (232.2)	0.95 (1.38)	88.40	C-Mo-T _x	8
	231.4 (234.8)	1.70 (1.89)	6.71	Mo ⁺⁵	8
	232.8 (235.6)	1.30 (1.45)	4.89	Mo ⁺⁶	8
Ti 2p _{3/2} (Ti 2p _{1/2})	455.1 (460.8)	1.13 (1.20)	60.89	Ti-C	8
	456.1 (461.5)	1.36 (1.50)	20.22	Ti ⁺² -C	8
	457.6 (462.2)	1.70 (1.80)	9.68	Ti ⁺³ -C	8
	459.3 (463.5)	1.60 (1.80)	9.21	TiO ₂	8
C 1s	282.3	1.22	31.31	C-Mo	5
	284.8	1.27	33.73	C-C	5
	285.2	1.33	19.35	CH _x	5
	286.3	1.31	15.12	C-O	5
	288.4	1.00	0.49	-COO	5
O 1s	530.0	1.13	44.68	MoO _x	9
	530.9	1.08	19.27	Mo-O _x	9
	532.0	1.63	22.23	Mo-OH	9

	532.7	1.16	7.50	C-O	9
	533.7	1.53	6.32	H ₂ O	9
F 1s	684.6	2.2	16.61	C-Mo-F _x	10
	687.0	1.84	19.10	AlF _x	10
	689.8	1.79	65.29	C-F	10

LB-Mo ₂ Ti ₂ C ₃ T _x	BE (eV)	FWHM (eV)	Fraction	Assigned to	Ref.
Mo 3d _{5/2} (Mo 3d _{3/2})	229.3 (232.4)	1.01 (1.34)	90.51	C-Mo-T _x	8
	231.6 (235.0)	1.70 (1.89)	4.80	Mo ⁺⁵	8
	233.1 (235.8)	1.30 (1.45)	4.69	Mo ⁺⁶	8
Ti 2p _{3/2} (Ti 2p _{1/2})	455.3 (461.1)	1.13 (1.20)	51.80	Ti-C	8
	456.1 (461.8)	1.50 (1.66)	28.39	Ti ⁺² -C	8
	458.0 (462.5)	1.70 (1.80)	9.70	Ti ⁺³ -C	8
	459.5 (463.7)	1.60 (1.80)	10.11	TiO ₂	8
C 1s	282.6	1.22	23.03	C-Mo-T _x	5
	284.8	1.18	46.87	C-C	5
	285.3	1.00	13.12	CH _x	5
	286.4	1.46	16.33	C-O	5
	288.5	0.50	0.65	-COO	5
O 1s	530.3	1.26	36.64	MoO _x	9
	531.2	0.90	10.18	Mo-O _x	9
	531.8	0.82	9.02	Mo-OH	9
	532.6	1.56	37.64	C-O	9
	533.7	1.12	6.52	H ₂ O	9
F 1s	685.0	2.12	16.82	C-Mo-F _x	10
	687.3	2.50	19.45	AlF _x	10
	689.9	1.59	63.73	C-F	10
Br 3d _{5/2} (Br 3d _{3/2})	69.3 (70.1)	1.68 (1.68)	1.00	Mo-Br	11

1. Natu, V. et al. A critical analysis of the X-ray photoelectron spectra of $Ti_3C_2T_z$ MXenes. *Matter* **4**, 1224–1251 (2021).
2. Kamysbayev, V. et al. Covalent surface modifications and superconductivity of two-dimensional metal carbide MXenes. *Science* **369**, 979-983 (2020).
3. Jawaid, A. et al. Halogen etch of Ti_3AlC_2 MAX phase for MXene fabrication. *ACS Nano* **15**, 2771-2777 (2021).
4. Zhao, S. S. et al. Li-ion uptake and increase in interlayer spacing of Nb_4C_3 MXene. *Energy Stor. Mater.* **8**, 42-48 (2017).
5. Jayaweera, P. M., Quah, E. L. & Idriss, H. Photoreaction of ethanol on TiO_2 (110) single-crystal surface. *J. Phys. Chem. C* **111**, 1764-1769 (2007).
6. Yang, J. et al. Two-dimensional Nb-based M_4C_3 solid solutions (MXenes). *J. Am. Ceram. Soc.* **99**, 660-666 (2016).
7. Maya, L. Ammonolysis of niobium(V) bromide. *Inorg. Chem.* **26**, 1459-1462 (1987).
8. Gandla, D., Zhang, F. M. & Tan, D. Q. Advantage of larger interlayer spacing of a $Mo_2Ti_2C_3$ MXene free-standing film electrode toward an excellent performance supercapacitor in a binary ionic liquid-organic electrolyte. *ACS Omega* **7**, 7190-7198 (2022).
9. Yamamoto, S. et al. In situ x-ray photoelectron spectroscopy studies of water on metals and oxides at ambient conditions. *J. Phys.: Condens. Matter* **20**, 184025 (2008).
10. Halim, J. et al. Synthesis and Characterization of 2D Molybdenum Carbide (MXene). *Adv. Funct. Mater.* **26**, 3118–3127 (2016).
11. Kumar, P. et al. Hexamolybdenum clusters supported on graphene oxide: Visible-light induced photocatalytic reduction of carbon dioxide into methanol. *Carbon* **94**, 91-100 (2015).

Comment 3. Why is there FWHM change for the same components before and after -Br substitution?

Response: The XPS curves are different before and after -Br substitution. In certain elements and component of XPS curves vary greatly, so there is no guarantee that the FWHM is exactly the same. This is consistent with the observation in literature (*Science* 2020, 369, 979-983).

Comment 4. The authors need to fit the MXene contributions in XPS with asymmetric peaks, using symmetric peaks causes unwanted extra components. For ex: in the case of Mo 3d peak in Fig S14 and d, the contribution from oxide will be much lower than shown. See some recent reviews on XPS of MXenes, they discuss this exact problem.

Response: We would like to thank for the reviewer's suggestion. We carefully analyzed the XPS based on recent reviews of MXenes.

Action: We fit the MXene contribution in XPS with asymmetric peaks in Supplementary Information.

Fig. R1. High resolution XPS spectra of (a, d) Mo 3d, (b, e) Ti 2p, (c, f) O 1s, (g, h) C 1s and (i) F 1s of $\text{Mo}_2\text{Ti}_2\text{C}_3\text{T}_x$ and $\text{LB-Mo}_2\text{Ti}_2\text{C}_3\text{T}_x$. (Fig. S14 in revised manuscript)

Comment 5. Please quantify the chemical formulas based on XPS before and after salt treatment.

Response: We have added the chemical formulas in Supplementary Information.

Action: We show the chemical formula after Lewis basic halide treatment in Supplementary Information.

“The chemical formula of LB-Ti₃C₂T_x is Ti₃C_{1.62}O_{1.3}OH_{1.1}F_{1.1}Br_{0.6} based on XPS.”

Comment 6. The authors say " At the same time, the desolvated cations could intercalate into the narrow interlayer space of MXenes to efficiently increase the distance". Ions always intercalate between MXene layers to balance charge. So there is nothing special per say here that the authors are emphasizing. De-solvated ions can be achieved in regular MXene by mere vacuum drying, and even in this case, will the desolvated ions not solvate as soon as they come in contact with the electrolyte in supercapacitors?

Response: We agree with the reviewer that the desolvated ions intercalate between MXene layers to balance charge after -Br termination substitution. To make it clear, we change the related statement.

Action: According to reviewer's comment, we have modified and marked the sentence in our revised manuscript, page 3:

“At the same time, the desolvated cations intercalate between MXene layers to balance charge after surface termination substitution.”

Comment 7. "For short conclusion, the desolvated anions (Br⁻) in AlBr₃/NaBr/KBr play an important role on substituting termination and desolvated cations (Na⁺ and K⁺) could simultaneously enlarge the interlayer spacing." I think the authors are missing out on the point that both, the ion and the -Br contribute to increased interlayer spacing. Also, I still don't get what's so special about naked ion intercalation.

Response: We agree with the reviewer that -Br termination is the main contribute to increased interlayer spacing.

Action: We have modified the sentence in our revised manuscript, page 7:

“For short conclusion, the desolvated ions in AlBr₃/NaBr/KBr play an important role on substituting termination and enlarging the interlayer spacing.”

Comment 8. In response to Comment 5, the authors need to compare their d-spacings with MXene synthesised in a mix of HI+HCl or HBr+HCl for true comparison. There are few papers in the literature that do that.

Response: Thanks for reviewer's comments. We have added the related data in the revised manuscript.

Action: According to the reviewer's suggestion, we added the interlayer spacing of MXene synthesised in a mix of HBr+HCl in **Table R4-4**.

Table R4-4. The interlayer spacing of multilayer MXene after intercalation. (Table S1 in revised manuscript)

Intercalating solution	(Multilayer MXene) Interlayer spacing	Ref.
LiF/HCl	13.50-14.00 Å	1
LiF/HCl	12.42 Å	2
MgCl ₂	14.29 Å	3
NaCl	11.85 Å	3
N-butyllithium	12.38 Å	4
LiOH	12.38 Å	4
LiOH	13.60 Å	5
SnCl ₄	12.42 Å	6
LiCl	11.50 Å	6
NaCl	11.01 Å	6
N ₂ H ₄ ·H ₂ O	12.70 Å	7
KOH	12.50 Å	8
HBr/HCl	13.50 Å	9
LiF/HCl	12.90 Å	This work
NaBr, KBr	14.60 Å	This work

Reference

1. Ghidui, M., Lukatskaya, M. R., Zhao, M. Q., Gogotsi, Y. & Barsoum, M. W. Conductive two-dimensional titanium carbide 'clay' with high volumetric capacitance. *Nature* **516**, 78-81 (2014).

2. Lipatov, A. et al. Effect of synthesis on quality, electronic properties and environmental stability of individual monolayer Ti_3C_2 MXene flakes. *Adv. Electron. Mater.* **2**, 1600255 (2016).
3. Al-Temimy, A. et al. Impact of cation intercalation on the electronic structure of $\text{Ti}_3\text{C}_2\text{T}_x$ MXenes in sulfuric acid. *ACS Appl. Mater. Interfaces* **12**, 15087-15094 (2020).
4. Chen, X. F. et al. N-butyllithium-treated $\text{Ti}_3\text{C}_2\text{T}_x$ MXene with excellent pseudocapacitor performance. *ACS Nano* **13**, 9449-9456 (2019).
5. Wang, H. B. et al. Achieving high-rate capacitance of multi-layer titanium carbide (MXene) by liquid-phase exfoliation through Li-intercalation. *Electrochem. Commun.* **81**, 48-51 (2017).
6. Hu, A. L., Y, J., Zhao, H. Z., Zhang, H., Li, W. One-step synthesis for cations intercalation of two-dimensional carbide crystal Ti_3C_2 MXene. *Appl. Surf. Sci.* **505**, 144538 (2020).
7. Mashtalir, O. et al. Effect of hydrazine intercalation on structure and capacitance of 2D titanium carbide (MXene). *Nanoscale* **8**, 9128-9133 (2016).
8. Li, J. et al. Achieving high pseudocapacitance of 2D titanium carbide (MXene) by cation intercalation and surface modification. *Adv. Energy Mater.* **7**, 1602725 (2017).
9. Anion adsorption, $\text{Ti}_3\text{C}_2\text{T}_z$ MXene multilayers, and their effect on claylike swelling. *J. Phys. Chem. C* **122**, 23172–23179 (2018).

Comment 9. The authors say " Numerous efforts have been devoted to the regulation of $\text{Ti}_3\text{C}_2\text{T}_x$ MXene, however, tuning interlayer spacing and surface halogen termination of other MXenes (besides $\text{Ti}_3\text{C}_2\text{T}_x$) remains challenging while demanded". I disagree with the fact that its challenging. It is for most part lack of motivation to try what works for Ti_3C_2 as is on other MXenes when there is no apparent benefit of using them over Ti_3C_2 .

Response: Thanks for the reviewer's comments and totally understand the concern. We think developing the synthesis method for other MXenes is meaningful, as it may help to explore new properties and advance the potential applications of MXene family. Accordingly, we have changed the related statement.

Action: We have modified the sentence in our revised manuscript, page 1:

“Numerous efforts have been devoted to the regulation of $\text{Ti}_3\text{C}_2\text{T}_x$ MXene, however, tuning interlayer spacing and surface halogen termination of other MXenes (besides $\text{Ti}_3\text{C}_2\text{T}_x$) is rarely reported while demanded.”

REVIEWER COMMENTS

Reviewer #4 (Remarks to the Author):

The authors have answered all my comments.

One last query that I have based on their latest round of revision is:

The authors say the chemical formula they determined using XPS is $\text{Ti}_3\text{C}_1.62\text{O}_1.3\text{OH}_1.1\text{F}_1.1\text{Br}_0.6$. If one adds the total number of moles of terminations it adds up to 4.1. But for MXenes it is always seen that the terminations occupy the FCC sites on the top of the surface Ti layers. So based on that, it is not possible to exceed the moles of total termination species more than 2. The authors should justify why their numbers are double than what is generally observed or make corrections in their XPS.

Response Notes

(Simultaneously tuning interlayer spacing and termination of MXenes by Lewis-basic halides)

We appreciate for the editor and reviewers' constructive suggestions to our manuscript. All the suggestions are very helpful for us to improve our research. We have addressed all comments of the reviewers, and our responses to the reviewer's comments are listed below point-by-point. The revised sections have been highlighted in the revised manuscript.

Reviewer #4

The authors say the chemical formula they determined using XPS is $\text{Ti}_3\text{C}_{1.6}\text{O}_{1.3}\text{OH}_{1.1}\text{F}_{1.1}\text{Br}_{0.6}$. If one adds the total number of moles of terminations it adds up to 4.1. But for MXenes it is always seen that the terminations occupy the FCC sites on the top of the surface Ti layers. So based on that, it is not possible to exceed the moles of total termination species more than 2. The authors should justify why their numbers are double than what is generally observed or make corrections in their XPS.

Response: We thank the reviewer for pointing this out. We have re-fitted the XPS data and re-calculate the chemical formula for $\text{LB-Ti}_3\text{C}_2\text{T}_x$ in Supplementary Information. The recalculated chemical formula is $\text{Ti}_3\text{C}_{1.6}\text{O}_{0.5}(\text{OH})_{0.4}\text{F}_{0.8}\text{Br}_{0.6}$ based on XPS. The number of moles at the end of $\text{LB-Ti}_3\text{C}_2\text{T}_x$ recalculated formula adds up to 2.3, which is close to 2. It is reasonable for the molar value of termination to fluctuate around 2 (*Matter* 2021, **4**, 1224-1251, *Nat. Mater.* 2021, **19**, 894-899).

Action: We have added the chemical formula for $\text{LB-Ti}_3\text{C}_2\text{T}_x$ in Supplementary Information.

“The chemical formula of $\text{LB-Ti}_3\text{C}_2\text{T}_x$ is $\text{Ti}_3\text{C}_{1.6}\text{O}_{0.5}(\text{OH})_{0.4}\text{F}_{0.8}\text{Br}_{0.6}$ based on XPS.”

Supplementary Figure 5. High resolution XPS spectra of (a, d) Ti 2p, (b, e) C 1s and (c, f) O 1s of $Ti_3C_2T_x$ and LB- $Ti_3C_2T_x$.